# *circZNF827* nucleates a transcription inhibitory complex to balance neuronal differentiation

Anne Kruse Hollensen[1], Henriette Sylvain Thomsen[1], Marta Lloret-Llinares[1,2], Andreas Bjerregaard Kamstrup[1], Jacob Malte Jensen[3], Majbritt Luckmann[1], Nanna Birkmose[1], Johan Palmfeldt[4], Torben Heick Jensen[1], Thomas B Hansen[1], Christian Kroun Damgaard[1]*

[1]Department of Molecular Biology and Genetics, Aarhus University, Aarhus, Denmark; [2]European Bioinformatics Institute (EMBL-EBI), European Molecular Biology Laboratory, Wellcome Genome Campus, Hinxton, Cambridge, United Kingdom; [3]Bioinformatics Research Centre, Aarhus University, Aarhus, Denmark; [4]Department of Clinical Medicine, Research Unit for Molecular Medicine, Aarhus University, Aarhus, Denmark

**Abstract** Circular RNAs are important for many cellular processes but their mechanisms of action remain poorly understood. Here, we map circRNA inventories of mouse embryonic stem cells, neuronal progenitor cells and differentiated neurons and identify hundreds of highly expressed circRNAs. By screening several candidate circRNAs for a potential function in neuronal differentiation, we find that *circZNF827* represses expression of key neuronal markers, suggesting that this molecule negatively regulates neuronal differentiation. Among 760 tested genes linked to known neuronal pathways, knockdown of *circZNF827* deregulates expression of numerous genes including nerve growth factor receptor (*NGFR*), which becomes transcriptionally upregulated to enhance NGF signaling. We identify a *circZNF827*-nucleated transcription-repressive complex containing hnRNP-K/L proteins and show that knockdown of these factors strongly augments NGFR regulation. Finally, we show that the ZNF827 protein is part of the mRNP complex, suggesting a functional co-evolution of a circRNA and the protein encoded by its linear pre-mRNA host.

*For correspondence:
ckd@mbg.au.dk

## Introduction

The mammalian non-coding transcriptome, which includes long noncoding RNAs (lncRNAs) and circular RNAs (circRNAs), plays pivotal roles in biological decisions during differentiation and normal cell maintenance (reviewed in *Chekulaeva and Rajewsky, 2019*; *Deveson et al., 2017*; *Kopp and Mendell, 2018*). Even though circRNAs were already identified several decades ago (*Capel et al., 1993*; *Kos et al., 1986*; *Nigro et al., 1991*; *Sanger et al., 1976*), they only recently have emerged as a large class of abundant noncoding RNAs that exhibit cell-type- and tissue-specific expression patterns (*Ashwal-Fluss et al., 2014*; *Hansen et al., 2013*; *Jeck et al., 2013*; *Memczak et al., 2013*; *Rybak-Wolf et al., 2015*; *Salzman et al., 2013*; *Salzman et al., 2012*) (reviewed in *Chekulaeva and Rajewsky, 2019*; *Ebbesen et al., 2017*; *Salzman, 2016*). CircRNAs are generated by the canonical spliceosome in a non-linear backsplicing fashion (*Cocquerelle et al., 1993*; *Jeck et al., 2013*; *Memczak et al., 2013*; *Pasman et al., 1996*; *Salzman et al., 2012*). During circRNA biogenesis, flanking intronic sequences are thought to bring splice sites within critically close proximity, either by direct basepairing between inverted repeats (e.g. Alu-repeats) or facilitated by interactions between flanking intron-bound RNA-binding proteins (RBPs) (*Ashwal-Fluss et al., 2014*;

*Conn et al., 2015*; *Ebbesen et al., 2017*). Most circRNAs are primarily localized to the cell cytoplasm (*Ashwal-Fluss et al., 2014*; *Hansen et al., 2013*; *Jeck et al., 2013*; *Memczak et al., 2013*; *Rybak-Wolf et al., 2015*; *Salzman et al., 2012*), and recent evidence suggests that nuclear export of circRNAs in human cells is influenced by the size of the given molecules, where larger circRNAs (>800 nucleotides) are dependent on DExH/D-box helicase UAP56 (DDX39B), whereas smaller species are dependent on URH49 (DDX39A) (*Huang et al., 2018*).

Several reports have provided evidence that circRNAs play important roles in various fundamental cellular processes. Well-described examples are the *CDR1as/ciRS-7* and *Sry* circRNAs that function to negatively regulate *Mir7-1* and *Mir138-1* activity, respectively, by sequestration (miRNA sponging), leading to increased mRNA expression of their respective miRNA-targets (*Hansen et al., 2013*; *Memczak et al., 2013*). However, it has also been suggested that the majority of circRNAs are likely not bona fide miRNA sponges, simply due to relatively low copy numbers and a low number of miRNA-binding sites per molecule, leaving efficient miRNA regulation ambigous in many cases (*Chekulaeva and Rajewsky, 2019*; *Ebbesen et al., 2017*). Examples of circRNAs acting as binding scaffolds for RBPs, or RBP sponges, which in turn affect their canonical function in for example premRNA splicing and protein translation, have been reported (*Abdelmohsen et al., 2017*; *Ashwal-Fluss et al., 2014*). Nuclear variants coined exon-intron circular RNAs (ElciRNAs), have, due to their retention of intronic sequences, been shown to promote transcription by recruitment of U1 snRNP to transcription units by a not fully clarified mechanism (*Li et al., 2015*). Many abundant circRNAs originate from the 5' end of their precursor transcripts, often giving rise to backsplicing into parts of the 5'UTR of their linear relative (*Jeck et al., 2013*; *Memczak et al., 2013*; *Rybak-Wolf et al., 2015*). The prevalence af these AUG circRNAs suggests that at least a subset of circRNAs could have protein-coding potential via a cap-independent translation mechanism (*Stagsted et al., 2019*). This is consistent with both early studies of Internal Ribosome Entry Sites (IRES) placed in a circRNA context (*Chen and Sarnow, 1995*), as well as more recent studies reporting examples of translation-competent circRNAs (*AbouHaidar et al., 2014*; *Legnini et al., 2017*; *Pamudurti et al., 2017*; *Yang et al., 2017*). However, global analyses of hundreds of ribosome profiling and mass-spec datasets, suggests that these few examples are specialized events, and not a generally applicable function of circRNAs (*Stagsted et al., 2019*).

RNA-sequencing of RNA isolated from mouse and human tissues along with various cell lines suggests that circRNAs are most abundantly expressed in the brain, compared to other tissues and that circRNAs are particular enriched in neuronal synaptosomes (*Rybak-Wolf et al., 2015*). In line with this, cells derived from both embryonal carcinoma (P19) and neuroblastoma (SHSY-5Y) subjected to neuronal/glial differentiation show tightly regulated circRNA expression profiles during neuronal development, that include upregulation of numerous common circRNAs (*Rybak-Wolf et al., 2015*). Piwecka et al., demonstrated that a *ciRS-7* knockout mouse displayed downregulated *Mir7-1* levels, alterations in sensorimotor gating associated with neuropsychiatric disease and abnormal synaptic transmission, suggesting that *ciRS-7* and *Mir7-1* are important for normal brain function in the mouse (*Piwecka et al., 2017*). Adding to the complexity of this regulatory network, a long noncoding RNA (lncRNA), *Oip5os1* (also coined *Cyrano*), promotes the destruction of *Mir7-1*, which in turn upregulates *ciRS-7* by a still unidentified mechanism (*Kleaveland et al., 2018*). One circRNA, *circSlc45a4*, which is very abundant in the cortex of the mouse and human brain, has recently been shown to negatively regulate neuronal differentiation, both in cell cultures and in developing mice, where its knockdown dysregulates the balance between specialized cortex neurons by unknown molecular mechanisms (*Suenkel et al., 2020*).

Despite these intricate molecular interactions between circRNA, miRNA, and lncRNA, many important questions regarding neuronal differentiation and function remain unanswered. For example, it is largely unknown how the tightly controlled expression of circRNAs potentially affects neuronal development. Here, we present the circRNA inventory of mouse embryonic stem cells (mESC), neuronal progenitor cells (NPC) and differentiated glutamatergic neurons, which represents a well-established model for CNS-type neuronal differentiation (*Bibel et al., 2007*). We report thousands of RNase R-resistant circRNAs of which many are differentially regulated during neuronal development. In a screen for circRNA function using an established human model for neuronal differentiation, we identify *circZNF827* as a negative regulator of neuronal differentiation. Although being almost exclusively localized to the cell cytoplasm, the nuclear population of this circRNA impacts several genes of relevance in neuronal differentiation, at the level of transcription, including nerve

growth factor receptor (*NGFR*), which becomes robustly upregulated upon *circZNF827* knockdown. Mechanistically, our evidence suggests that *circZNF827* is a necessary scaffold for a transcription-repressive complex containing its own host-encoded protein; ZNF827, together with hnRNP K and hnRNP L.

## Results

### The circRNA profile of mESCs changes markedly upon neuronal differentiation

To determine whether circRNAs can influence neuronal differentiation, we initially mapped the circRNA inventories at different stages of neuronal differentiation and compared these to other available circRNA datasets of neuronal origin from mice and humans (*Rybak-Wolf et al., 2015*). Identification of circRNAs from RNA-seq experiments has often been based on quantification of relatively few reads across the circRNA backsplicing junction (circBase *Glažar et al., 2014*), and current circRNA prediction algorithms inevitably lead to the calling of false positives (*Hansen et al., 2016*; *Jeck and Sharpless, 2014*). Hence, to immediately validate the circular nature of *to-be* called circRNAs, we first performed standard rRNA depletion and subsequently either included or excluded RNase R treatment step prior to RNA-sequencing. Specifically, we used an established differentiation model for CNS-type glutamatergic neurons, based on E14 mouse embryonic stem cells (mESCs) that reportedly yields a purity of glutamatergic neurons of >90% (*Bibel et al., 2007*). RNA was isolated from three stages of differentiation, mESCs, neuronal progenitor cells (mNPCs) or neuronal differentiation day 8 (mN8) and rRNA depleted (+/- RNase R) prior to library preparation and RNA-seq (*Figure 1A*). Successful differentiation at the mNPC and mN8 stages was confirmed by the appearance of elongated intercellular dendritic extensions (mN8) (*Figure 1—figure supplement 1A*) and robust upregulation of several classical neuronal markers including, *Ntrk2*, *Map2*, and *Tubb3* (mNPC and mN8), while stem cell pluripotency marker *Nanog* became significantly reduced upon differentiation (*Figure 1—figure supplement 1B*). Using available circRNA prediction tools CIRI2 (*Gao et al., 2018*), find_circ (*Memczak et al., 2013*) and CIRCexplorer2 (*Zhang et al., 2016*) on the non-RNase R-treated RNA, we identified between 792–1167 circRNAs in mESC, 2230–2893 circRNAs in NPC and 1902–2316 circRNAs in differentiated neurons at mN8 stage (*Figure 1B*). Upon RNase R treatment most circRNAs either remained unchanged or became enriched, but a considerable fraction of the predicted circRNAs in mESC, mNPC, and mN8 preparations, became depleted by the 3′−5′ exonuclease (CIRCexplorer2: 19.5–36.5%; CIRI2: 7.2–16.6%; find_circ: 38.7–52.3% depleted) (*Figure 1—figure supplement 1C*). All prediction algorithms showed a correlation between expression level and RNase R resistance, suggesting that mostly low-count circRNAs candidates are likely false positives (*Figure 1—figure supplement 1D*). From a total of 3581 enriched circRNAs after RNase R treatment (all stages), 1449 circRNAs overlapped between all three circRNA prediction algorithms, and this subset represents a high-confidence circRNA inventory (*Hansen et al., 2016*; *Figure 1C* and *Supplementary file 1*). We next assessed the *circular-to-linear* ratio of identified circRNAs (find_circ), by comparing splice site usage in circular vs. linear splicing events (*Memczak et al., 2013*; *Rybak-Wolf et al., 2015*). This analysis revealed vast differences in the steady-state levels of these isoforms and demonstrated that many circRNA species are considerably more abundant than their linear precursors (*Figure 1D*). Confirming previous results (*Rybak-Wolf et al., 2015*), introns flanking the circRNAs are generally longer than average introns and circRNAs often tend to cluster at the 5′ end of their respective precursor RNA (*Figure 1—figure supplement 1E–F*). Our results suggest that in order to obtain high confidence circRNA inventories from RNA-seq data, it is beneficial to use multiple circRNA prediction algorithms and to enrich for bona fide circRNAs, by depletion of linear RNAs using RNase R.

We next tested differential circRNA expression during differentiation, which revealed marked changes in circRNA expression over the 16-day timecourse (*Figure 1E*; left panel). Kmeans clustering of circRNAs by expression (Top 100 highest expressed) pattern showed two main clusters with peak expression at mNPC and mN8 (*Figure 1E*; right panel). Comparison with previously identified mouse and human homologue circRNAs, isolated from mouse brain regions or cell lines of either murine or human origin (*Rybak-Wolf et al., 2015*), revealed a substantial overlap between circRNAs at differentiated stages (e.g. 80% of all 1449 circRNAs found in differentiated murine p19 cells and primary

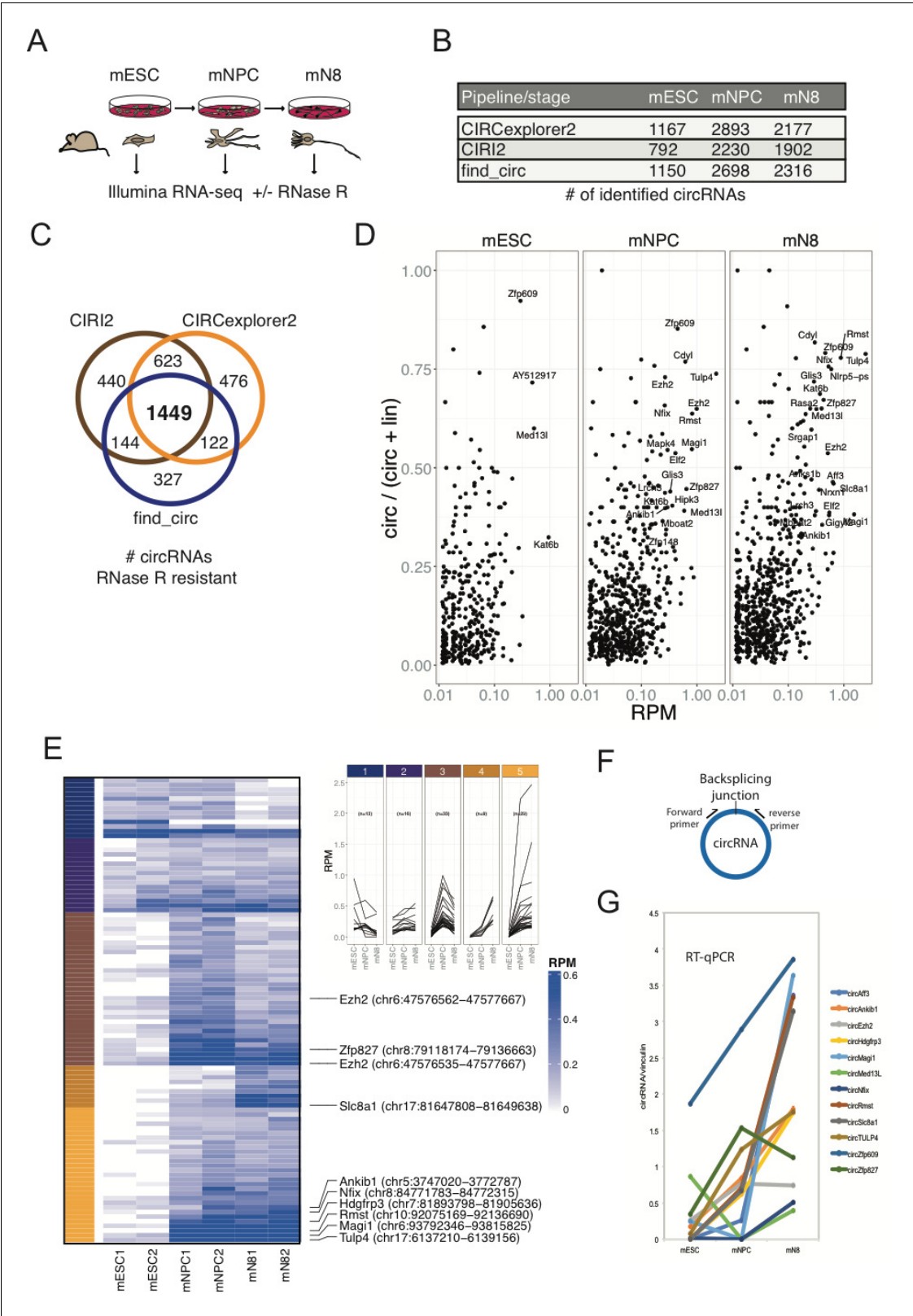

**Figure 1.** Determining the circRNA inventories of mESC, NPC and differentiated glutamatergic neurons and their differential regulation. (**A**) Schematic illustration of workflow for differentiation and RNA-seq. (**B**) Number of circRNAs detected by indicated circRNA prediction algorithm in different stages. (**C**) Venn-diagram showing 1449 common circRNAs of a total of 3581 circRNAs predicted by the different algorithms (as indicated next to the diagram) that are either constant or enriched upon RNase R treatment. (**D**) circRNA/circRNA+linear precursor ratios as a function of expression level (RPM) at the

*Figure 1 continued on next page*

Figure 1 continued

three sequenced stages. (E) Left: Heatmap showing differential expression of top-100 expressed circRNAs (RPM scale to the right), with selected examples of circRNAs as indicated along with genomic coordinates (mm10). Top: K-means analysis displaying five different expression profiles during differentiation (same color code given to the left of the heatmap). (F) circRNA RT-qPCR strategy spanning the backsplicing junction. (G) RT-qPCR validaton of selected circRNAs. Data are depicted as mean ± SD (two biological replicates).

The online version of this article includes the following figure supplement(s) for figure 1:

**Figure supplement 1.** Mouse neuronal cell morphology, expression pattern of select markers and characteristics of circRNA inventory.

---

neurons, 45% of Top100 found in human SH-SY5Y and 75% overlap with circRNAs found in the human ENCODE data previously analyzed *Rybak-Wolf et al., 2015*; *Stagsted et al., 2019*; *Figure 1—figure supplement 1G*). We confirmed differential expression of a subset of the most abundant and upregulated circRNAs (*circTulp4*, *circMagi1*, *circRmst*, *circEzh2*, *circHdgfrp3*, *circZfp827*, *circMed13l*, *circZfp609*, *circSlc8a1*, *circNfix*) using RT-qPCR with amplicons across the backsplicing junction (*Figure 1F–G*). 75% of the top-100 expressed mouse circRNAs was also found in human circRNA datasets (*Rybak-Wolf et al., 2015*; *Figure 1—figure supplement 1G*). We conclude that significant changes in circRNA expression patterns are induced upon neuronal differentiation and that the majority of these circRNAs are conserved between various neuronal cell-types originating from humans and the mouse.

## Knockdown of circZNF827 stimulates neuronal marker expression

To ascertain whether the highly upregulated circRNAs might contribute to the process of neuronal differentiation, we next depleted a number of candidate circRNAs by RNA interference. We first tested knockdown efficiency of *circZfp827* (*circZNF827* in humans) by lentivirally delivered dishRNAs (*Kaadt et al., 2019*) targeting the backsplicing junction in either mESC, p19, SH-SY5Y or L-AN-5 cells, of which the latter three cell lines are well-established models of neuronal differentiation following retinoic acid treatment. Knockdown efficiency in mESC and p19 proved relatively poor (30–60% remaining circRNA) compared to the two human cell lines: SH-SY5Y (10% remaining) and L-AN-5, which displayed superior results (<8% remaining) (*Figure 2A* and *Figure 2—figure supplement 1A*). Moreover, when testing SH-SY5Y cells for an increase of neuronal differentiation markers *NTRK2*, *NEFL*, *MAP2* and *TUBB3* upon retinoic acid treatment, only *NTRK2* was significantly upregulated upon differentiation (*Figure 2—figure supplement 1B*), whereas these genes showed a more expected and dynamic expression pattern in L-AN-5 cells (*Figure 2B*). We therefore transduced L-AN-5 cells with lentiviral dishRNA vectors to perform knockdown of 14 candidate circRNAs (*Figure 2A* and *Figure 2—figure supplement 1C*; *circTULP4*, *circSLC8A1*, *circZNF609*, *circHDGFRP3*, *circMAGI*, *circRMST*, *circZNF827*, *circANKIB1*, *circMED13L*, *circCDYL*, *circUNC79*, *circHIPK3*, *circNFIX*, *circCAMSAP1*) (*Supplementary file 2a*) and subsequently subjected these to retinoic acid-induced differentiation followed by neuronal marker quantification in order to probe for changes in differentiation. In general, we observed efficient knockdown (*Figure 2A* and *Figure 2—figure supplement 1C*). While the majority of knockdowns did not significantly change neuronal marker expression, knockdown of *circZNF827* (and to a lesser extent *circANKIB1*), produced a significant and reproducible increase in neuronal marker expression upon differentiation (*Figure 2B* and *Figure 2—figure supplement 2A*). Importantly, the linear *ZNF827* mRNA was not affected by backsplicing junction-specific knockdown (*Figure 2—figure supplement 2B*). The upregulation of neuronal markers following *circZNF827* knockdown was also evident at the protein level for *MAP2* and *TUBB3* (*Figure 2C*, and quantified to the right). In addition, proliferation assays demonstrated a smaller S-phase population (32% to 24%) upon *circZNF827* knockdown, suggesting lowered replication kinetics (*Figure 2D* and *Figure 2—figure supplement 3A–B*). This phenomenon was accompanied by a minor stall in $G_2/M$ phase, while $G_0/G_1$ phase was not significantly affected between control and *circZNF827* knockdown. Taken together, our results suggest that *circZNF827* knockdown exerts a repressive effect on proliferation, while it enhances neuronal marker expression and hence differentiation.

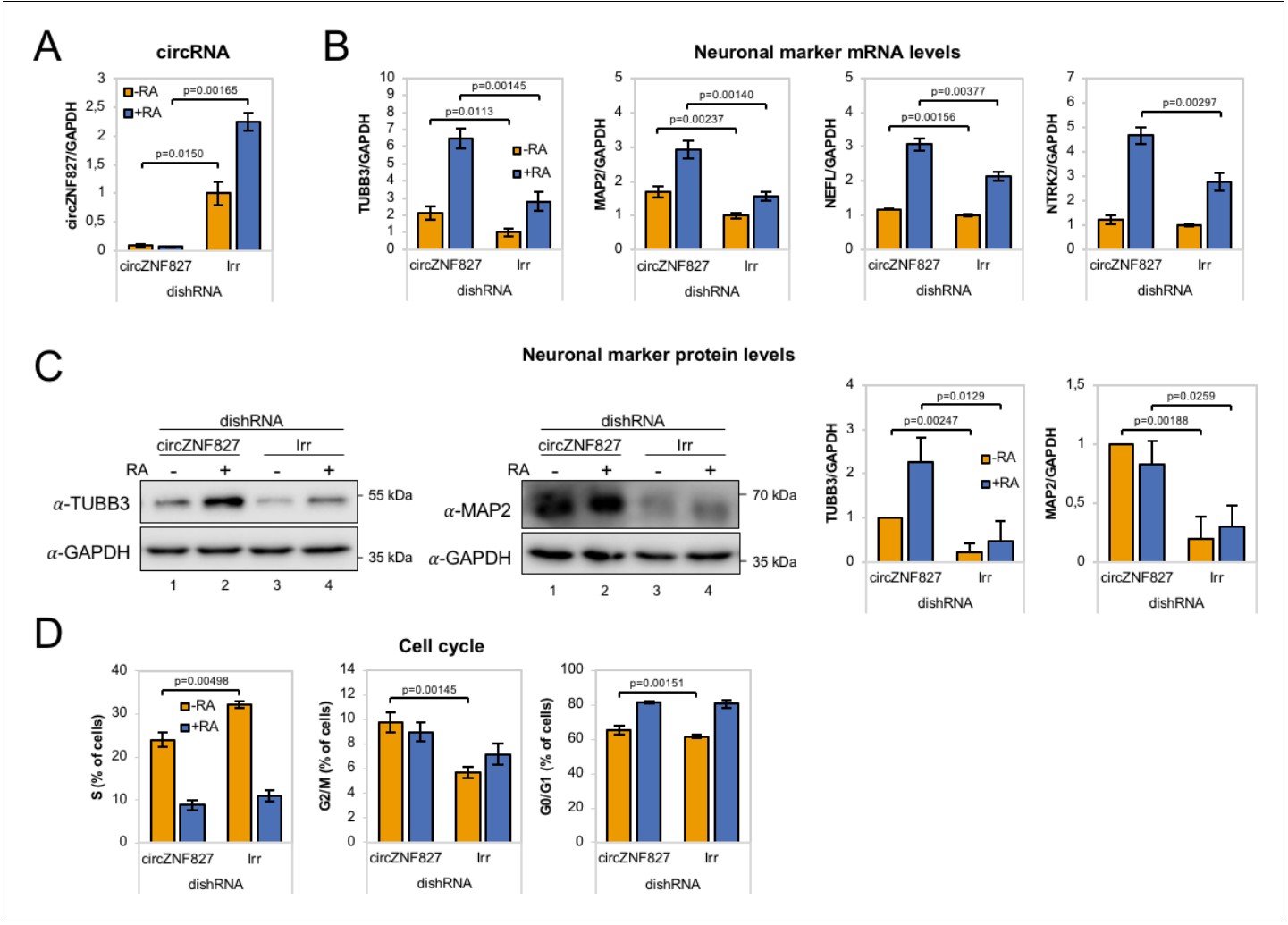

**Figure 2.** *circZNF827* regulates neuronal marker expression levels. (**A**) RT-qPCR analysis evaluating knockdown of *circZNF827* with dicer-independent short hairpin RNAs (dishRNAs) in the neuroblastoma cell line L-AN-5. (**B**) Relative mRNA levels of the neuronal markers *TUBB3*, *MAP2*, *NEFL*, and *NTRK2* evaluated by RT-qPCR upon knockdown of *circZNF827*. The mRNA expression levels were evaluated by RT-qPCR after 4 days of RA-mediated neuronal differentiation. (**C**) Western blotting (left panels) of TUBB3 and MAP2 upon *circZNF827* knockdown. GAPDH was used as loading control. The results of quantification of band intensities from western blots are shown the right panels. One representative western blot and the quantification of three is shown. (**D**) Cell cycle assay based on flow cytometric measurements of EdU incorporation into newly synthesized DNA in L-AN-5 cells upon *circZNF827* knockdown. +RA: differentiated L-AN-5 cells. -RA: undifferentiated L-AN-5 cells. Irr: Irrelevant dishRNA. In all panels, data are depicted as mean ± SD (three biological replicates). p-Values were determined by a two-tailed Student's t test.

The online version of this article includes the following figure supplement(s) for figure 2:

**Figure supplement 1.** Evaluation of circZfp827/circZNF827 knockdown in different cell types and its impact on select neuronal marker expression.

**Figure supplement 2.** mRNA levels of neuronal markers upon circRNA knockdown.

**Figure supplement 3.** Flow cytometric gating strategy and raw data from cell cycle assay.

## circZNF827 controls retinoic acid receptor homeostasis

We next asked whether the retinoic acid receptors (RARs), which represent central nodes in relaying anti-proliferative differentiation cues during neuronal development (*Gudas and Wagner, 2011*), and are key targets of retinoic acid (RA), also become upregulated upon knockdown of *circZNF827*. Indeed, knockdown of *circZNF827* leads to a moderate but significant increased expression (1.5–2.5 fold) of *RARB* and *RARG*, while *RARB* remained constant (*Figure 3A*). Since most circRNAs have been reported to predominantly localize in the cell cytoplasm, we addressed the localization of *circZNF827*, *circANKIB1* and *circTULP4* by cellular fractionation. These circRNAs are mainly cytoplasmically localized in L-AN-5 cells (~90% cytoplasmic signal) (*Figure 3B*). We therefore hypothesized

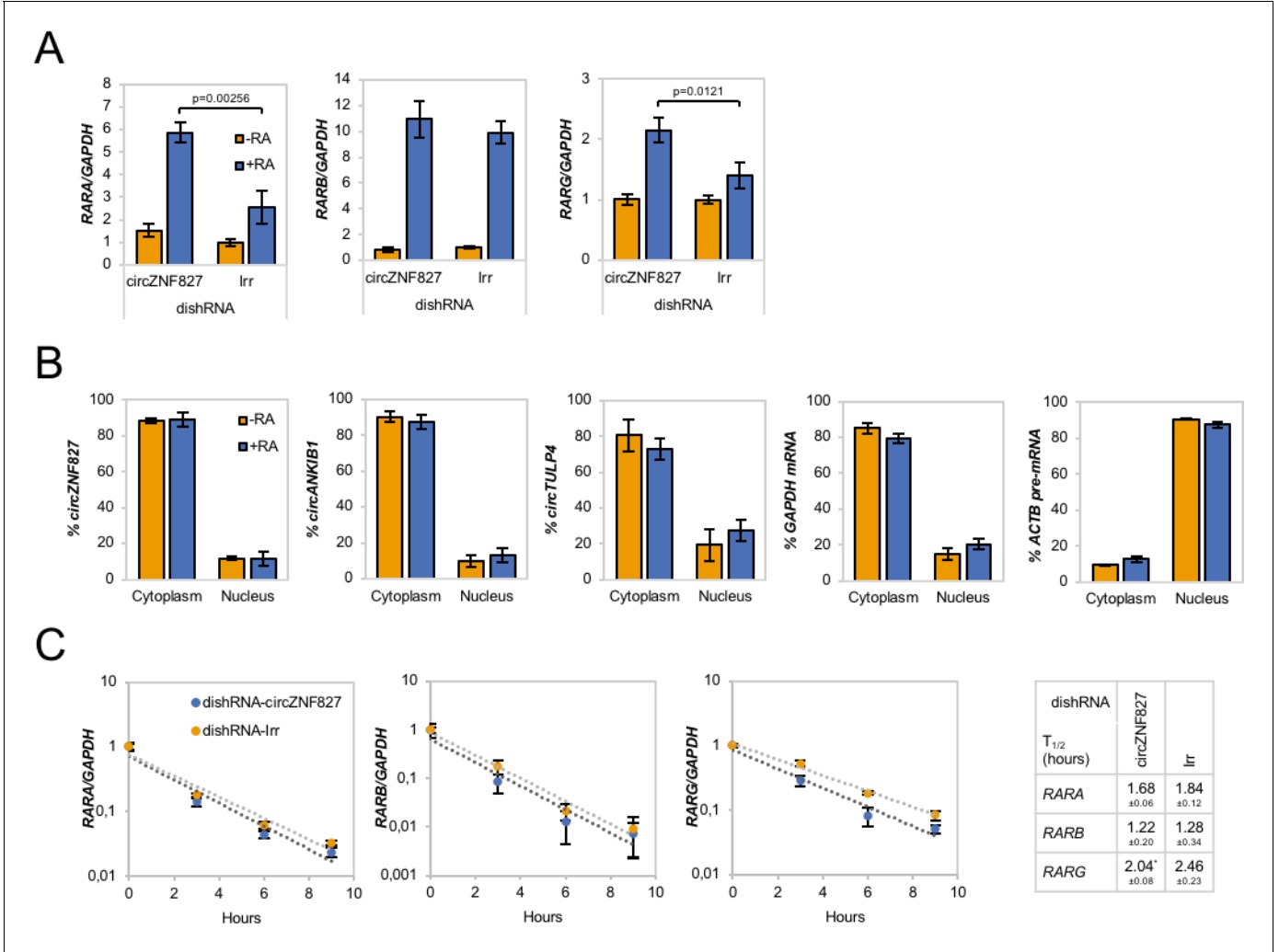

**Figure 3.** Increased *RAR* expression upon *circZNF827* knockdown. (**A**) mRNA expression levels of the RAR receptors *RARA*, *RARB*, and *RARG* in L-AN-5 cells upon *circZNF827* knockdown evaluated by RT-qPCR. (**B**) Subcellular localization of the circRNAs *circZNF827*, *circANKIB1*, and circTULP4 examined by RT-qPCR after fractionation of differentiated L-AN-5 cells into cytoplasmic and nuclear fractions. *GAPDH* mRNA and *ACTB* pre-mRNA levels was used for validation of the purity of the cytoplasmic and nuclear fractions. (**C**) BrU pulse-chase mRNA decay assay evaluating decay rates of *RAR* mRNAs upon *circZNF827* knockdown. The *RAR* mRNA expression levels were evaluated by RT-qPCR. In right panel, half-lives of the *RARs* obtained in the experiment are indicated. +RA: differentiated L-AN-5 cells. -RA: undifferentiated L-AN-5 cells. Irr: Irrelevant dishRNA. In all panels data are depicted as mean ± SD (three biological replicates). p-Values were determined by a two-tailed Student's t test.

The online version of this article includes the following figure supplement(s) for figure 3:

**Figure supplement 1.** *RAR* mRNA transcription rates estimated after BrU-labeling of newly synthesized RNA in differentiated L-AN-5 cells by RT-qPCR.

that *circZNF827* could potentially affect *RAR*-mRNA stability post-transcriptionally in the cell cytoplasm. However, BrU pulse-chase mRNA decay assays demonstrated no significant change in *RAR*-mRNA decay rates upon knockdown of *circZNF827* (*Figure 3C*). Next, we investigated transcription rates, by treating cells with a short pulse of BrU, followed by BrU immunoprecipitation to quantify de novo labeled RNA, serving as a proxy for transcription rates during control- or knockdown of *circZNF827*. As expected from the constant mRNA decay rates, BrU incorporation was moderately upregulated, although significantly only for *RARA*, upon *circZNF827* knockdown (*Figure 3—figure supplement 1*). Although this moderate increase in transcription measured at a single late time-point, cannot readily explain the larger effect on the *RARA* mRNA level (2.5-fold) measured at steady-state, our results suggest that *circZNF827* contributes to controlling the RARA receptor levels in order to keep neuronal differentiation in check.

## circZNF827 knockdown affect multiple genes in neuronal signaling

Our results indicate that L-AN-5 cells are lowering their proliferation rates and promote RAR-signaling by transcriptional upregulation of one of these transcription factors when *circZNF827* levels are low. To test how *circZNF827* knockdown affects other key factors of the neuronal transcriptome, we next performed Nanostring analyses using a neuro-differentiation/pathology panel of 760 genes with RNA purified from differentiated or non-differentiated L-AN-5 cells. 135 genes become differentially expressed (9 upregulated and 126 downregulated, fold change > +/- 2, p<0.05) due to *circZNF827* knockdown after differentiation (*Figure 4A*, *Supplementary file 2b*). In line with a potential negative regulatory function of *circZNF827* on neuronal differentiation, GO-term analyses show enrichment of terms including axon/dendrite structure, neural cytoskeleton, transmitter synthesis, neural connectivity, growth factor signaling and trophic factors among differentially expressed genes (*Figure 4B*). The most significantly upregulated gene is *nerve growth factor receptor* (*NGFR*), which plays a central role in regulating neuronal differentiation, death, survival, and neurite outgrowth (*Yamashita et al., 1999*; *Zhu et al., 2012*). Conversely, *Phosphatase and tensin homolog* (*PTEN*), *STAT3* and *NAD(P)H quinone dehydrogenase 1* (*NQO1*) were all significantly downregulated upon *circZNF827* knockdown (2–4 fold), which reportedly also contributes positively to neuronal differentiation (*Lyu et al., 2015*; *Ma et al., 2017*), and in case of the latter, also renders cells more susceptible to energetic and proteotoxic stress (*Hyun et al., 2012*). Since *NGFR* is a key regulator of neuronal differentiation and the highest upregulated gene upon *circZNF827* knockdown, we next focused on the mechanism of its upregulation. Using both RT-qPCR and western blotting, demonstrated a strong upregulation at both the protein and mRNA level (*Figure 4C* and *Figure 4—figure supplement 1*). This upregulation was not due to changes in mRNA decay rates, since BrU pulsechase mRNA decay assays yielded nearly identical mRNA half-lives upon *circZNF827* knockdown (*Figure 4D*). To address whether the observed changes in gene expression are elicited at the transcriptional level, we subjected cells to a short BrU-pulse prior to BrU immunoprecipitation and Nanostring hybridization. Interestingly, *NGFR* and also *ATP8A2* proved to be highly upregulated (~4–6 fold) at the level of transcription (*Figure 4E–F*), while only NQO1 and not PTEN and STAT3 exhibited significantly reduced transcription activity (ranging from ~1.3- to ~4-fold) (*Figure 4E*). Also, the MAP2 gene did not change its de novo RNA output, suggesting that the tuning of the steady-state levels of *PTEN*, *STAT3* and *MAP2* mRNAs, as initially observed (*Figures 2B* and *4A*), are mainly facilitated by posttranscriptional changes to mRNA stability. If *circZNF827* is involved in a *direct* transcription-associated complex that regulates *NGFR* output, the transcriptional effects elicited by *circZNF827* knockdown would require nuclear knockdown of the circRNA. Indeed, the use of dicerindependent shRNA (dishRNA) vectors proved very efficient in depleting nuclear circRNA (*Figure 4G*).

Next, we assayed the cellular impact of *NGFR* upregulation upon *circZNF827* knockdown. To this end, we NGF-treated L-AN-5 cells subjected to either control or *circZNF827* knockdown, and quantified downstream signaling output by quantification of *FOS*, which is a well-known downstream 'immediate early' target of *NGFR* signaling. *FOS* levels increased significantly, strongly indicating that the higher levels of NGFR protein indeed leads to functional increase in NGFR signaling (*Figure 4H* and *Figure 4—figure supplement 1*), which can at least in part explain the upregulation of neuronal markers. Taken together, we conclude that *circZNF827* serves to keep neuronal differentiation 'in check' by limiting expression of, and signaling by, *RARA* and *NGFR*.

## circZNF827 interacts with transcriptional regulators hnRNP K and -L

To address the mechanism by which these transcriptional and post-transcriptional events are controlled by *circZNF827*, we next sought to identify its protein interactome. To this end, we synthesized biotin-labeled *circZNF827* (linear version) and control RNAs (*circTULP4*, *circZNF609*, *circHDGFRP3* and *circSLC8A1*) in vitro and subjected these to pull-down experiments using L-AN-5 cell lysates and streptavidin-coupled magnetic beads as previously described (*Seitz et al., 2017*). Silver-stained SDS-PAGE gels of retained proteins revealed unique profiles, suggesting that specific proteins exhibited increased affinity toward *circZNF827*, although prominent RNA-binding proteins common to both control RNAs and *circZNF827* could also be observed (*Figure 5—figure supplement 1A*). By subjecting pulled-down fractions to LC-MS/MS, we identified several *circZNF827*-specific proteins, including hnRNP K and -L, while others (e.g. DHX9 and DDX3X) bound strongly to any

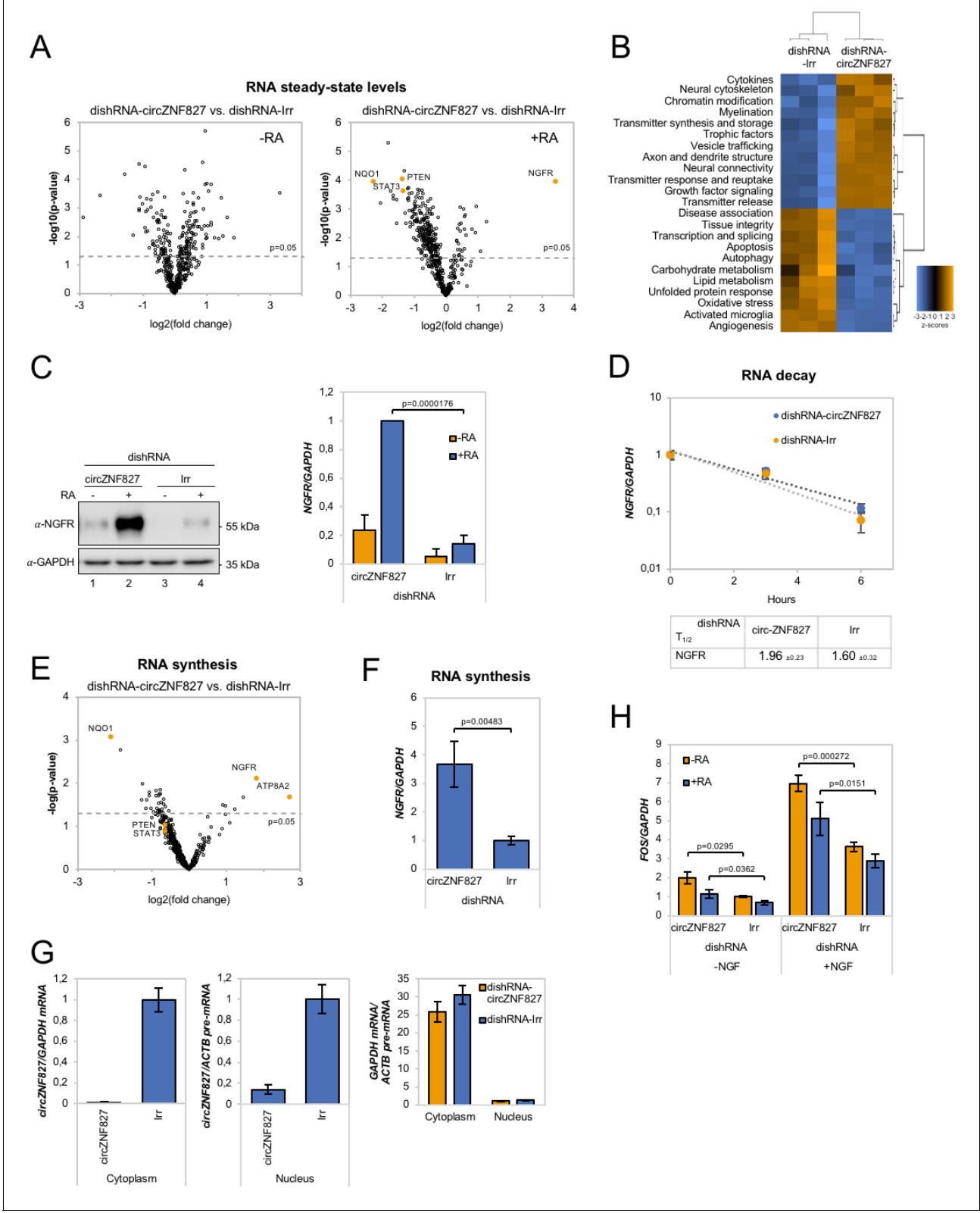

**Figure 4.** *circZNF827* regulates NGFR expression. (**A**) Volcano plot based on a Nanostring analysis of the expression of ~800 neuropathology-related genes upon *circZNF827* knockdown in L-AN-5 cells vs control without RA treatment (left panel) or with RA treatment (right panel). (**B**) GO-term analysis based on genes found differentially expressed by the Nanostring analysis upon *circZNF827* knockdown in differentiated L-AN-5 cells. (**C**) Western blotting (left panel) of NGFR upon *circZNF827* knockdown in L-AN-5 cells. GAPDH was used as loading control. The result of quantification of band

*Figure 4 continued on next page*

*Figure 4 continued*

intensities from the western blots is shown in the right panel. One representative western blot and the quantification of three is shown. (D) BrU pulse-chase mRNA decay assay evaluating decay rates of *NGFR* mRNAs upon *circZNF827* knockdown. In the bottom panel, the half-lives of NGFR obtained in the experiment are indicated. (E) Volcano plot showing mRNAs with changed synthesis rates estimated after BrU-labeling of newly synthesized RNA by Nanostring analysis using the neuropathology panel. (F) RT-qPCR-based validation of the Nanostring analysis shown in (E). (G) Evaluation of *circZNF827* knockdown in L-AN-5 cells after subcellular fractionation into nuclear and cytoplasmic RNA fractions by RT-qPCR. *GAPDH* mRNA and *ACTB* pre-mRNA levels was used for validation of the purity of the cytoplasmic and nuclear fractions. (H) *FOS* mRNA levels evaluated by RT-qPCR after *circZNF827* knockdown and NGF stimulation of L-AN-5 cells. +RA: differentiated L-AN-5 cells. -RA: undifferentiated L-AN-5 cells. Irr: Irrelevant dishRNA. Data are depicted as mean ± SD (three biological replicates). (D–F) One representative western blot is shown. p-Values were determined by a two-tailed Student's t test.

The online version of this article includes the following figure supplement(s) for figure 4:

**Figure supplement 1.** *NGFR* mRNA expression upon *circZNF827* knockdown in NGF-stimulated L-AN-5 cells.

of the bait RNAs (*Figure 5A*). To validate these interactions, we performed RNA-immunoprecipitaiton (RIP) using monoclonal anti-hnRNP K or -L antibodies followed by qRT-PCR across the backsplicing junction, and observed a significant enrichment of *circZNF827* compared to IgG controls (>100 fold), suggesting that these interactions can be recapitulated in L-AN-5 cells (*Figure 5B*). As expected for these highly expressed RNA-binding proteins, both proteins associate with *GAPDH* mRNA, but in the case of hnRNP L, the IP/input ratios were ~18 fold higher for *circZNF827*, whereas hnRNP K displayed a similar enrichment of *GAPDH* mRNA as of *circZNF827* (*Figure 5B*). Scrutinizing the *circZNF827* sequence for putative-binding sites for hnRNP K and -L using eCLIP datasets (ENCODE consortium), proved unfeasible due to low expression levels of the *ZNF827* gene in the K562 and HepG2 cells used by ENCODE. Using RBPmap (*Paz et al., 2014*), which is based on established RBP consensus binding sequences, revealed a potential high-affinity cluster for primarily hnRNP L binding and one site for hnRNP K in the most 3' part of the circle-encoding sequence (*Figure 5—figure supplement 1B–C*). According to *circZNF827* secondary RNA structures predicted by Mfold, these binding sites are located into mostly single-stranded regions within the circRNA, consistent with the binding preferences of most hnRNP proteins toward single-stranded RNA (*Figure 5—figure supplement 1D*). To further characterize these interactions, we prepared a stable HEK293 Flp-In T-rex cell-line expressing *circZNF827* (*Figure 5—figure supplement 1E–F*) under the control of a tetracycline-inducible promoter (tet-on), based on the laccase vector system (*Kramer et al., 2015*). We then performed RIP by immunoprecipitation of endogenous hnRNP K or -L and observed a remarkable enrichment of exogenous *circZNF827*, compared to control IgG or *GAPDH* mRNA (*Figure 5C and E*). hnRNP L gave a particularly high IP/Input ratio (>200 fold enrichment over IgG), consistent with the results from: (1) the L-AN-5 RIP, (2) the pull-down LC-MS/MS experiment and (3) the prediction of several hnRNP L binding site clusters in *circZNF827*. To test whether the predicted hnRNP K/- L binding sites in the 3' part of *circZNF827* are indeed sites of interaction, we constructed a deletion mutant that removes the putative binding sites (*Figure 5—figure supplement 1C*, dark grey letters and *Figure 5—figure supplement 1D*, dotted dark grey line) and prepared HEK293 stable cell lines (*Figure 5—figure supplement 1E–F*). Although this mutant was expressed at somewhat lower levels than WT *circZNF827*, it remained virtually unbound by hnRNP K/- L similar to IgG pull-down efficiency (*Figure 5C*).

We conclude that both hnRNP K and -L can be found in complex with endogenous or exogenous *circZNF827* in both L-AN-5 and HEK293 Flp-in T-Rex cells, likely via high-affinity binding sites in the 3' part of the circular RNA.

## Increasing expression of circZNF827 induces distinct hnRNP K and -L nuclear foci

hnRNP K is a well-documented transcriptional regulator (*Moumen et al., 2005*; *Thompson et al., 2015*) that is reported to interact directly with hnRNP L and -U (*Havugimana et al., 2012*; *Kim et al., 2000*; *Wan et al., 2015*) and bind both DNA and RNA (*Tomonaga and Levens, 1995*). To assess interactions between hnRNP K, -L and -U, and their potential dependence on *circZNF827*, we performed co-immunoprecipitation experiments using FLAG-tagged hnRNP K, -L, and -U, and subsequently probed for their interaction with endogenous proteins (*Figure 5D*) in HEK293 Flp-in cells either overexpressing *circZNF827* or not. hnRNP K co-immunoprecipitates both hnRNP U and

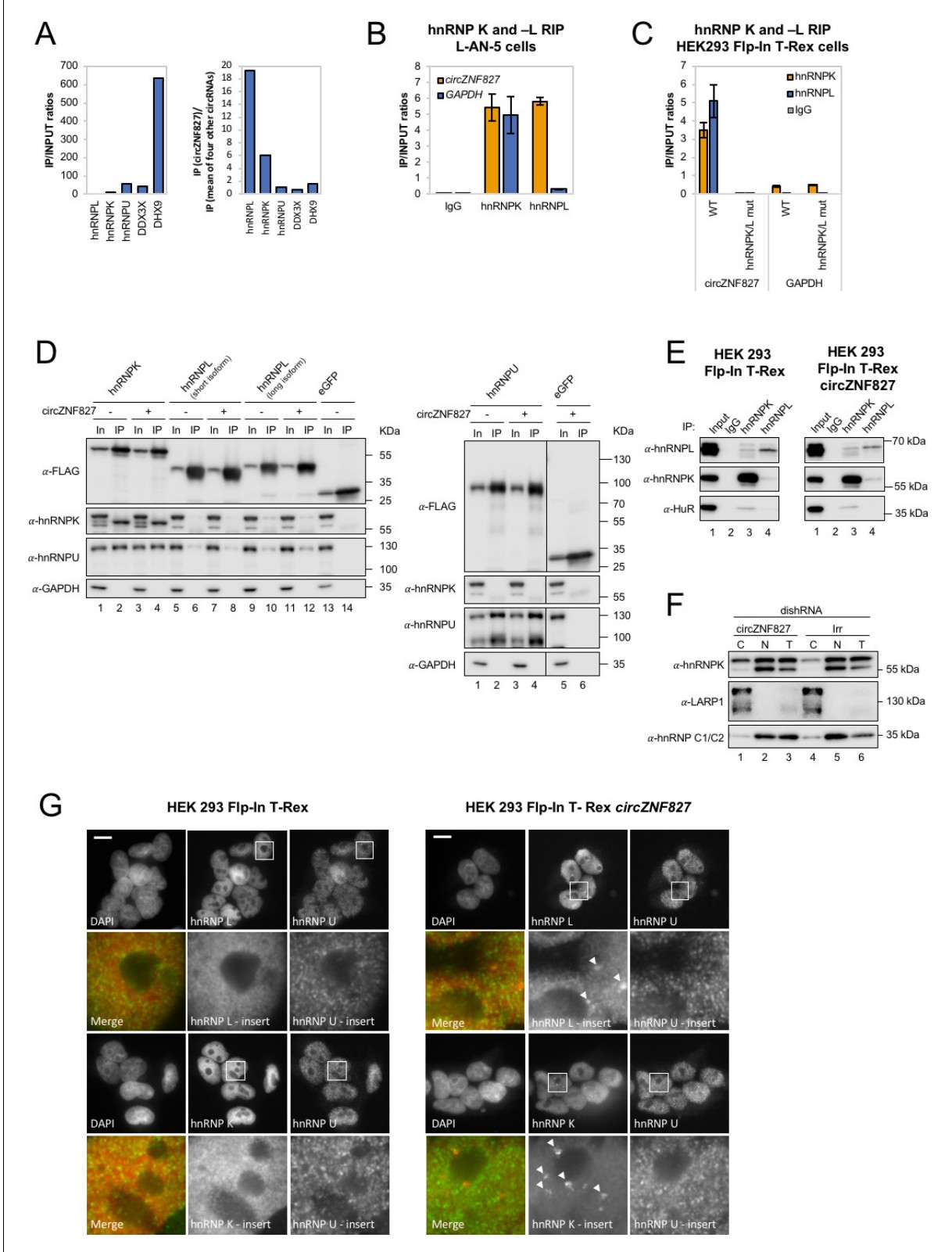

**Figure 5.** *circZNF827* interacts with and regulates the subcellular localization of hnRNP K and -L. (**A**) circRNA-RBP complex isolation from differentiated L-AN-5 cells followed by protein identification using mass spectrometry (LC-MS/MS). IP/Input ratios (based on IBAQ values) for selected RBPs (hnRNP L, hnRNP L, hnRNP U, DDX3X and DHX9) pulled down by *circZNF827* are shown in left panel. In the right panel IP ratios of selected RBPs pulled down by *circZNF827* relative to IP ratios for four other circRNAs (*circTULP4*, *circHDGFRP3*, *circSLC8A1*, and *circZNF609*) are shown. (**B**) RIP experiment evaluating *Figure 5 continued on next page*

*Figure 5 continued*

interaction between *circZNF827* and hnRNP K and -L in differentiated L-AN-5 cells. (**C**) RIP experiment evaluating interaction between both wildtype *circZNF827* (WT) and *circZNF827* with a deletion of predicted hnRNPK/L binding sites (hnRNPK/L mut) and hnRNP K and -L in the HEK293 Flp-In T-rex *circZNF827* cell lines. Co-immunoprecipitation (co-IP) of both exogenously FLAG-tagged (**D**) and endogenously (**E**) expressed hnRNP K, -L and -U in HEK293 Flp-In T-rex cells with and without *circZNF827* expression. GAPDH and HuR were used as loading controls in (**D**) and (**E**) respectively. (**F**) Western blot evaluating subcellular localization of hnRNP K in differentiated L-AN-5 cells upon *circZNF827* knockdown. LARP1 and hnRNP C1/C2 were used for validation of the purity of the cytoplasmic and nuclear fractions. (**G**) Co-immunofluorescence (co-IF) of hnRNP K, -L and -U in HEK293 Flp-In T-rex cells upon *circZNF827* overexpression. Arrows are pointing to hnRNP K- and hnRNP L-containing nuclear foci. Nuclei were visualized by DAPI staining. The scale bar indicates 10 µm. Irr: Irrelevant dishRNA. C: cytoplasmic fraction, N: nuclear fraction, T: total cell lysate. Data are depicted as mean ± SD (three biological replicates). (**B**), (**E**), and (**H**) One representative western blot is shown. Data shown in (**A**) and (**C**) are based on two and one replicates, respectively.

The online version of this article includes the following figure supplement(s) for figure 5:

**Figure supplement 1.** Mapping hnRNP K and hnRNP L binding sites within circZNF827.

hnRNP L (long isoform), but these interactions remain unaffected by increased expression of *circZNF827* (*Figure 5D*). In accordance with these findings, immunoprecipitation of endogenous hnRNP K and -L proteins in HEK293 Flp-in cells, confirmed that a hnRNP L/hnRNP K complex can be detected although only a small fraction of the total hnRNP K/- L populations co-immunoprecipitate the other (*Figure 5E*, left). This complex was not affected by overexpression of *circZNF827* (*Figure 5E*, right). Hence, exogenous *circZNF827* likely does not regulate bulk hnRNP K/L-complex assembly/disassembly in HEK293 cells per se.

To test whether *circZNF827* potentially regulates the normal subcellular distribution of hnRNP K, L-AN-5 cells were fractionated during control or *circZNF827* knockdown and lysates subjected to western blotting. We observe a small but significant and reproducible increase in the cytoplasmic population of hnRNP K upon *circZNF827* knockdown, suggesting that *circZNF827* retains, albeit a very small fraction of the hnRNP K population, in the nucleus (*Figure 5F*). To address this, we over-expressed *circZNF827* and monitored hnRNP K and L localization by immunofluorescence in HEK293 cells. Induction of *circZNF827* led to accumulation of hnRNP K and to a lesser extent hnRNP L in multiple distinct nuclear foci that were not detected in control cells (*Figure 5G*, white arrows). Taken together, our results suggest that while bulk hnRNP K and L complex formation is not affected by *circZNF827* levels, overexpression of the circRNA induces specific nuclear localization of hnRNP K and L.

## hnRNP K or -L knockdown increases NGFR levels

Could a *circZNF827*-dependent hnRNP K/L-containing nuclear complex regulate the output from the *NGFR* gene? If such a complex is instrumental in repressing *NGFR*, we predict that knockdown of any of these factors would enhance *NGFR* expression. In support of a role in hnRNP K-mediated regulation of *NGFR*, it was recently reported that hnRNP K knockdown strongly induces *NGFR* expression in mouse ES cells (*Thompson et al., 2015*). To test this in our context, we designed dishRNAs for hnRNP K and -L, transduced L-AN-5 cells and assayed for *NGFR* expression by qRT-PCR or Western blotting. Both knockdowns increased *NGFR* expression at both the mRNA and protein levels, similar to the effect of depleting *circZNF827* alone (*Figure 6A–B* and *Figure 6—figure supplement 1A–B*). However, co-depletion of *circZNF827* with any of these factors strongly augmented *NGFR* expression (four- to fivefold higher than individual knockdowns) (*Figure 6A–B*), suggesting that their effects are synergistic.

Given these results, a feasible possibility is that a hnRNP K/L-*circZNF827* complex could facilitate transcriptional repression of *NGFR* by interacting with gene-regulatory regions, consistent with *NGFR* upregulation upon *circZNF827* knockdown. To this end, publicly available ChIP-seq data (ENCODE consortium) in K562 and HepG2 cells demonstrate that hnRNP K indeed interacts with transcription regulatory regions (promoter proximal) of the *NGFR* gene (*Figure 6—figure supplement 1C*). To determine the *circZNF827*-dependence of a hnRNP K-containing complex that docks at the promoter region of *NGFR* gene in L-AN-5 cells, we next performed hnRNP K ChIP in the presence or absence of *circZNF827* and assayed for the *NGFR* promoter region by qPCR. Our results show that hnRNP K engagement at the NGFR promoter is decreased upon *circZNF827* knockdown compared to the *GAPDH* gene (*Figure 6C*), which displayed constant transcription rates in our

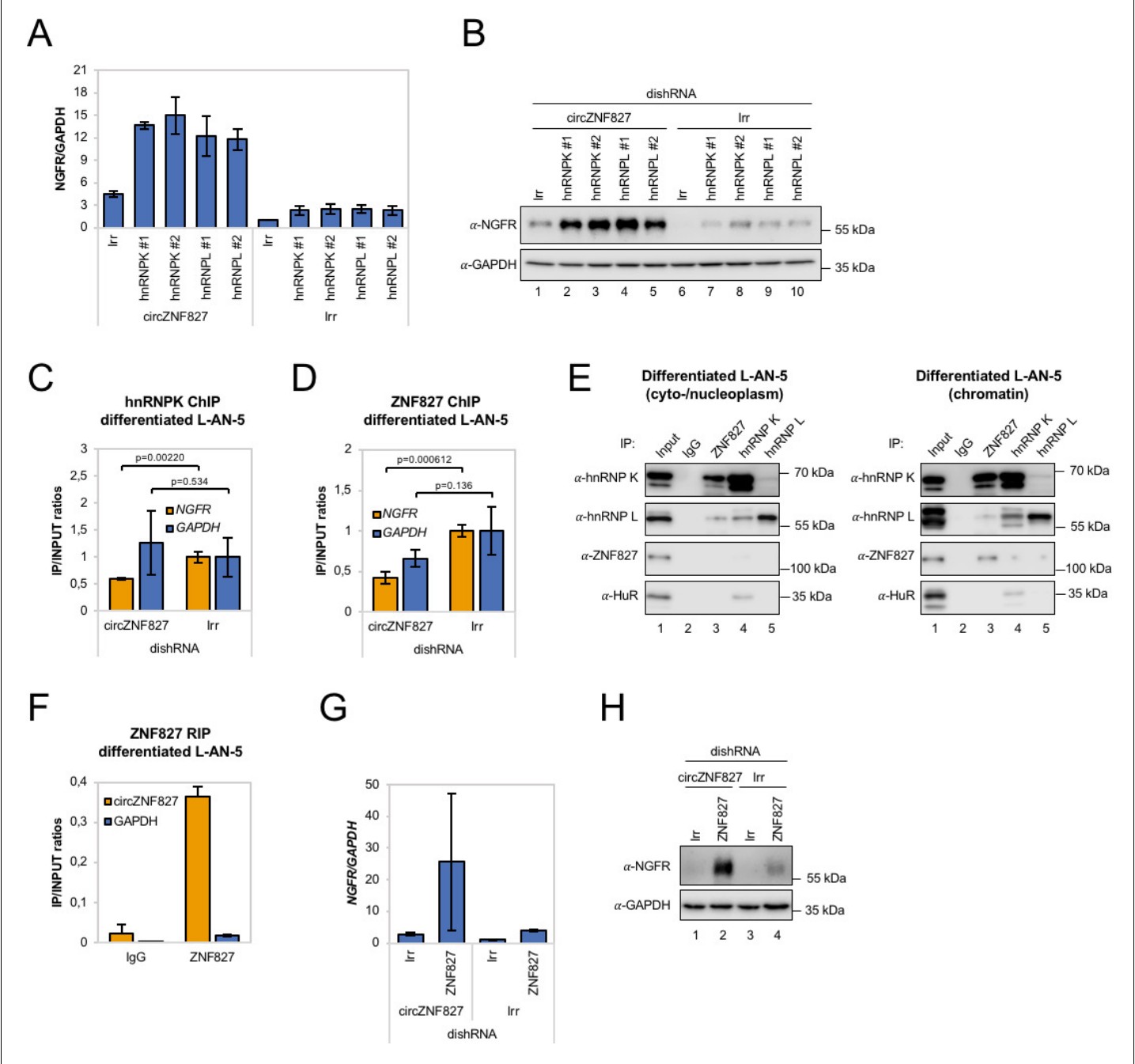

**Figure 6.** *circZNF827* regulates hnRNP K and ZNF827 activity in L-AN-5 cells. RT-qPCR (**A**) and western blotting (**B**) evaluating NGFR expression upon co-knockdown of *circZNF827* and either hnRNP K or -L in differentiated L-AN-5 cells. GAPDH was used as loading control for the western blots. #1 and #2: two different dishRNAs targeting the same RBP. ChIP experiment assessing association between the *NGFR* gene and hnRNP K (**C**) and ZNF827 (**D**) upon *circZNF827* knockdown in differentiated L-AN-5 cells. (**E**) Co-immunoprecipitation (co-IP) of ZNF827, hnRNP K, -L and ZNF827 in cyto-/nucleoplasm (left) or chromatin fractions (right; sonicated pellets from cleared lysates) of differentiated L-AN-5 cells. IgG was used as IP control. HuR was used as negative control. (**F**) RNA-immunoprecipitation of *circZNF827* by ZNF827 in differentiated L-AN-5 cells. RT-qPCR (**G**) and western blotting (**H**) evaluating NGFR expression upon co-knockdown of *circZNF827* and ZNF827 in differentiated L-AN-5 cells. GAPDH was used as loading control for the western blots. Irr: Irrelevant dishRNA. Data are depicted as mean ± SD (three biological replicates). p-Values were determined by a two-tailed Student's t test.

The online version of this article includes the following figure supplement(s) for figure 6:

**Figure supplement 1.** Knockdown and ChIP analyses of ZNF827 and hnRNP K/-L associated with results shown in *Figure 6A–B*.

previous BrU pulse labeling assay. We next wondered how this circRNP complex can interact with chromatin. It was recently demonstrated that hnRNP K partakes in a complex with chromatin-bound KRAB-domain zinc finger proteins (KRAB-ZNFs), and that hnRNP K is necessary for recruitment of a transcription inhibitory SETDB1/KAP1 complex, which catalyzes H3K27 trimethylation and hetero-chromatin formation (*Thompson et al., 2015*). We therefore hypothesized that ZNF827 protein, which does not harbor a discernible KRAB domain, could interact with either hnRNP K and/or its encoded circRNA and perhaps link this transcription repressive complex to the *NGFR* promoter. To test this hypothesis, we first conducted a ZNF827 ChIP experiment and probed for an interaction with the *NGFR* promoter region in the presence or absence of *circZNF827*. Indeed, our results suggest that ZNF827 interacts with the *NGFR* promoter and that the signal was significantly diminished upon *circZNF827* knockdown (*Figure 6D*). Interestingly, we observed a similar reduction in signal at the *RARA* promoter region, which we had previously found to be moderately, yet significantly, upregulated (*Figure 6—figure supplement 1D*). To address whether ZNF827 and hnRNP K interact, we next performed ZNF827 immunoprecipitation and found it to strongly associate with hnRNP K and to a lesser extent with hnRNP L in nucleoplasmic extracts (*Figure 6E*, left). Upon sonication of the remainder from the Triton X-100 extracted cleared lysates (chromatin enriched), we observed an even stronger association of hnRNP K with ZNF827, further suggesting that the complex is chromatin bound (*Figure 6E*, right). When assessing the ability of ZNF827 to interact with *circZNF827*, we observed a strong enrichment over IgG (~18 fold), and ZNF827 protein co-immunoprecipitated *circZNF827* more efficiently than *GAPDH* mRNA (~19 fold more enriched) (*Figure 6F*). Finally, we performed ZNF827 knockdown to test whether this would also augment expression of *NGFR* as observed upon hnRNP K/- L knockdown. Indeed, we observed a strong upregulation of *NGFR* upon ZNF827 knockdown, an effect that was further augmented in a *circZNF827* knockdown background (*Figure 6G–H* and *Figure 6—figure supplement 1E*).

Taken together, our results are consistent with a model where *circZNF827* represses *NGFR* transcription (and likely many other genes, including *RARA*) by bridging a hnRNP K/L-contaning inhibitory complex with their genomic loci, facilitated by the ZNF827 protein.

## circZNF827 is part of the chromatin-bound hnRNP-ZNF827 complex

Mechanistically, an important outstanding question is whether *circZNF827* nucleates the hnRNP-ZNF827 complex to potentially prepare/activate it for engagement with chromatin, or whether *circZNF827* itself, is part of the chromatin-bound complex. To address this question, we first cloned a *circZNF827* expression construct, which includes a sequence encoding 2 X MS2 hairpins (*Figure 7A*) and established stable inducible HEK293 cells (HEK293-*circZNF827*-MS2). Next, we transfected these cells, HEK293 control cells (Empty) or HEK293-*circZNF827*-WT (lacking MS2 hairpins) with a plasmid encoding FLAG-tagged MS2 coat protein (FLAG-MS2cp) and subsequently performed anti-FLAG ChIP analysis, to test whether the FLAG-tagged MS2cp becomes crosslinked to the NGFR promoter via *circZNF287*-2XMS2. The NGFR promoter signal was significantly higher in HEK293 cells harboring the *circZNF827*-2XMS2 expression cassette, when compared to both control cell lines and background signal on the GAPDH promoter remained constant in all three cell types (*Figure 7B*). Importantly, RIP experiments demonstrated that both hnRNP K and -L interact with *circZNF827*-2XMS2 with similar affinities as *circZNF827*-WT (*Figure 7—figure supplement 1*). These results demonstrate that *circZNF827* engages the *NGFR* promoter, suggesting that the circular RNA nucleates the hnRNP-ZNF827-containing complex on chromatin to limit *NGFR* expression (*Figure 7C*), which in turn contributes to an important balance between neuronal differentiation and self-renewal/proliferation.

## Discussion

Circular RNAs are by now considered as an important class of abundant and conserved RNAs but their functional potential has not been fully elucidated yet. Here, we identified high-confidence circRNA inventories of E14 mESCs, mNPCs and differentiated glutamatergic neurons, and show a generally high degree of conservation among circRNAs previously identified using cell lines and tissues of neuronal origin (*Rybak-Wolf et al., 2015*). Three different circRNA prediction pipelines, CIRI2 (*Gao et al., 2018*), find_circ (*Memczak et al., 2013*) and CIRCexplorer2 (*Zhang et al., 2016*), displayed marked differences in their predictions, which is in line with our earlier observations

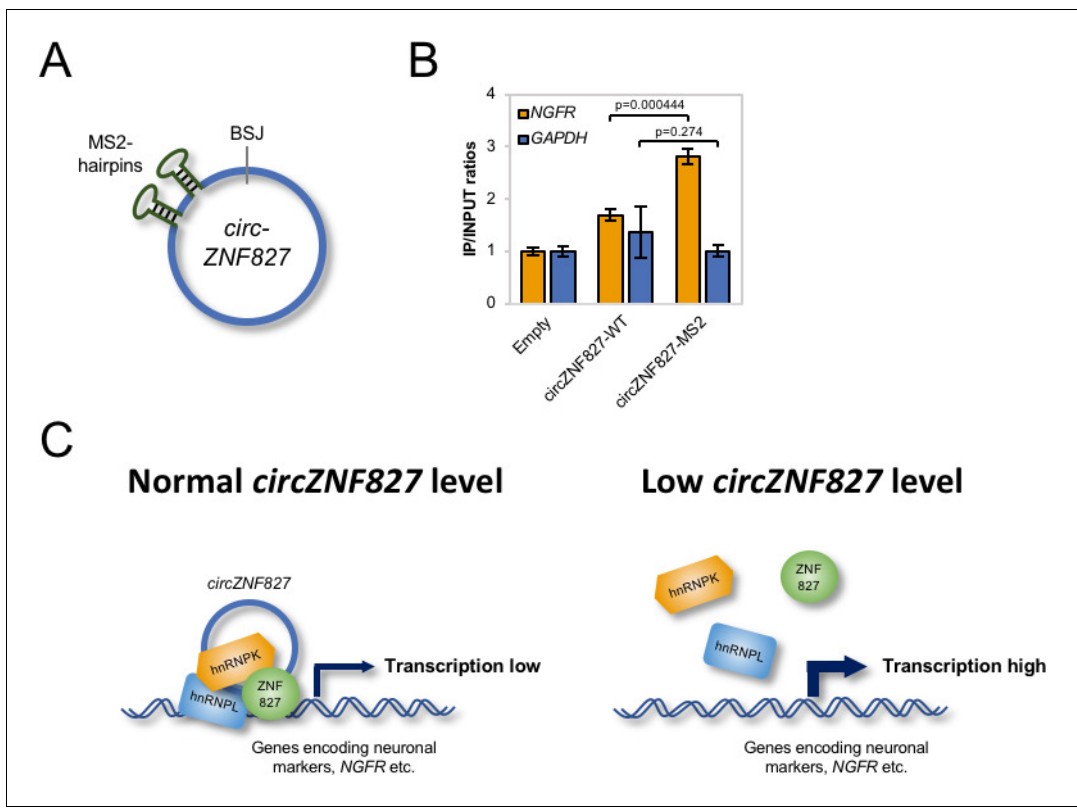

**Figure 7.** *circZNF827* is associated with the *NGFR* promoter region. (**A**) Schematic representation of *circZNF827* tagged with two MS2 hairpins. (**B**) ChIP experiment assessing association between the NGFR gene and *circZNF827* in HEK293 Flp-In T-rex cell lines expressing etiher wildtype *circZNF827* (*circZNF827*-WT) or MS2-tagged *circZNF827* (*circZNF827*-MS2). (**C**) Model illustrating how *circZNF827*, hnRNP K/- L and ZNF827 regulates target gene expression. Target genes, e.g. *NGFR* is bound by a transcription repressive complex consisting of *circZNF827*, hnRNP K, hnRNP L and ZNF827. High levels of *circZNF827*, induced by neuronal differentiation keeps further differentiation markers in check (left panel), while knockdown of *circZNF827* (or hnRNP K/L or ZNF827) allows for higher transcription rates of target neuronal marker genes including *NGFR*. Data in (**C**) are depicted as mean ± SD (three biological replicates).

The online version of this article includes the following figure supplement(s) for figure 7:

**Figure supplement 1.** hnRNP K and - L interact with both wildtype and MS2-hairpin-modifed circZNF827.

---

(*Hansen et al., 2016*). This could indicate that many reported circRNAs are false positives, especially when expressed at low levels. A surprisingly large fraction of initially called circRNAs by the three pipelines becomes depleted upon RNase R treatment (between 7.2% and 52.3%), with CIRI2 clearly being the most robustly performing predictor in terms of RNase R resistance. Among 3581 RNase R-resistant circRNAs, only 1449 were called by all three algorithms, suggesting that caution should be taken when predicting circRNAs from RNA-seq data and that including multiple prediction algorithms and/or an RNase R step prior to RNA-seq is beneficial.

Analyzing circRNA expression over the three neuronal developmental stages, we identified 116 differentially expressed circRNAs (>2 fold change). Of 14 tested circRNA candidates, knockdown of *circZNF827* in human L-AN-5 cells had a significant and positive impact on the expression of several classical neuronal markers, suggesting that the circRNA normally exerts a negative role in neuronal differentiation. Among 760 genes important to neuronal differentiation and disease, we found that *NGFR* was most strongly induced, also at the protein level, upon *circZNF827* knockdown. NGFR is a member of the TNF superfamily of receptors and relays, along with three paralogous receptor tyrosine kinases (NTRK1, NTRK2 and NTRK3), signals from the four mammalian neurotrophins (Nerve Growth Factor (NGF)), brain-derived neurotrophic factor (BDNF), neurotrophin-3 (NTF3) and neurotrophin 4 (NTF4) [*Bothwell, 2016*]. The regulation and functional output from the neurotrophins and

their receptors, which are interdependent proteins, is very complex and involves a multitude of effector proteins and interaction partners (*Bothwell, 2016*). NGFR can, depending on expression levels of the other neurotrophin receptors and their ligands, either induce death- or survival signaling to promote neuronal differentiation and control axonal growth or apoptosis (*Bothwell, 2016*). Whether *NGFR* upregulation is instrumental and causal for the enhanced expression of *NTRK2*, *NEFL*, *TUBB3*, and *MAP2* that we observe in the L-AN-5 neuroblastoma system, remains to be investigated. However, we did observe strongly augmented *FOS* expression (immediate early gene) upon treatment of L-AN-5 cells with NGF, when circZNF827 was downregulated, which suggests that NTRK1-mediated NGF response becomes enhanced by increased *NGFR* expression. It is possible that *NGFR* is induced to increase death-signaling, as a result of skewed and sub-optimal stoichiometry between key neuronal markers/effectors (e.g. *NTRK2*, *NEFL*, *TUBB3*, and *MAP2*). Such a scenario might be part of a normal surveillance system that monitors a strict and sequential appearance of differentiation factors; however, this awaits further disclosure in more physiologically relevant cell- and animal models.

Mechanistically, several lines of evidence support a model in which *circZNF827* plays a direct role in the transcriptional repression of the *NGFR* and *RARA* loci, potentially as a scaffolding-RNA for a hnRNP K/-L-ZNF827 containing complex (*Figure 7C*). *NGFR* mRNA decay rates remain unchanged upon *circZNF827* knockdown, while stready-state levels increase three- to fourfold. Knockdown of *circZNF287* resulted in significantly higher BrU incorporation rates in *NGFR* pre-mRNA and an increased association of PolII at the NGFR promoter. Importantly, hnRNP K and ZNF287 association with the NGFR promoter was decreased upon *circZNF287* knockdown. In addition, depletion of either hnRNP K, -L, or ZNF827, which all interact robustly with *circZNF827*, strongly augmented the transcriptional induction by *circZNF827* knockdown. We observed strong focal nuclear condensates containing endogenous hnRNP K and -L proteins in HEK293 Flp-in T-Rex cells stably expressing *circZNF827*. Although such condensates may be non-physiological entities (phase-separated hnRNPs), induced by high local concentrations of *circZNF827*, these results suggest that the circRNA could function as a scaffold that nucleates hnRNP K and -L, although not readily visible in the microscope when *circZNF827* levels are significantly lower.

It is well established that hnRNP K participates in transcriptional repression. hnRNP K can bridge classical DNA-binding KRAB-ZNF proteins and a KAP1/SETDB1-containing complex, which in turn facilitates heterochromatin formation – also in the *NGFR* gene of ES cells (*Thompson et al., 2015*). A similar mechanism was described by Huarte and collegues, where a p53-induced *lincRNA-p21* interacts with hnRNPK, which facilitates silencing of several downstream targets (*Huarte et al., 2010*). Interestingly, transcriptional stimulation, rather than repression, has been reported for intron-containing circRNAs (ElciRNAs), via recruitment of U1 snRNP to the transcriptional complex on their parental genes, which by definition requires exon-intron boundaries (*Li et al., 2015*). *CircZNF827* is a regular exonic circRNA, without intronic sequences (data not shown), perhaps explaining why it represses transcription as opposed to ElciRNAs. Another circRNA, *circSlc45a4*, was recently also shown to negatively regulate neuronal differentiation, both in cell cultures and in developing mice, where its knockdown dysregulates the balance between specialized cortex neurons (*Suenkel et al., 2020*). Taken together, our ChIP, RIP and co-IP results all suggest that the protein-product originating from the *circZNF827*-encoding pre-mRNA host, ZNF827, links hnRNP K/- L-*circZNF827* to chromatin, via its DNA-binding capacity, similar to known roles of KRAB-ZNF proteins. If so, *circZNF827* co-regulates target genes along with its precursor-encoded protein, which argues for co-evolutionary selection pressure to preserve both circRNA-generating and protein-coding sequences. *circMBNL1* has also been shown to regulate the activity of its cognate protein product (*Ashwal-Fluss et al., 2014*), suggesting that this phenomenon could be a common theme that awaits further investigation.

## Materials and methods

Sequences of all primers and probes used in the study are specified in *Supplementary file 2c-e*. Antibodies are described in *Supplementary file 2f*.

## Vector construction

To create plasmids for expression of dishRNAs, sense and antisense oligonucleotides were annealed and cloned into BglII/XhoI-digested pFRT/U6, resulting in vectors designated pFRT/U6-dishRNA (*Kaadt et al., 2019*). Subsequently, the U6-dishRNA expression cassettes were PCR-amplified from pFRT/U6-dishRNA vectors and inserted into ClaI/BsiWI-digested pCCL/PGK-eGFP-MCS (*Kaadt et al., 2019*). The resulting lentiviral transfer vectors were designated pCCL/U6-dishRNA-PGK-eGFP-MCS. As a negative control, an irrelevant dishRNA based on a previously described irrelevant shRNA (*Jakobsen et al., 2009*), which does not match any known sequence in the human genome was used throughout the study.

To generate plasmids for in vitro transcription of circRNAs, the exons encoding *circTULP4*, *circZNF827*, *circHDGFRP3*, *circZNF609*, and *circSLC8A1* were PCR-amplified from cDNA prepared from RNA isolated from L-AN-5 cells. PCR-amplicons encoding *circTULP4*, *circZNF827*, *circHDGFRP3*, and *circSLC8A1* were inserted in BamHI/NotI-digested pcDNA3/PL whereas PCR-amplicons encoding *circZNF609* were inserted in HindIII/NotI-digested pcDNA3/PL. The resulting plasmids were designated pcDNA3/circRNA. The plasmid for expression of the CTC lincRNA was constructed as previously described in *Seitz et al., 2017*.

For exogenous expression of *circZNF827*, the exons encoding *circZNF827* and *circZNF827*-hnRNPK/L mut were PCR-amplified from cDNA prepared from RNA isolated from L-AN-5 cells and inserted into PacI/SacII-digested pcDNA3.1(+)-Laccase2-MCS-exon-vector (*Kramer et al., 2015*). Subsequently, the Laccase-*circZNF827* and Laccase2-*circZNF827*-hnRNPK/L mut expression cassettes was inserted into HindIII/NotI- or BamHI/XhoI-digested pcDNA5_FRT/TO resulting in vectors designated pcDNA5_FRT/TO-Laccase2-*circZNF827* and -*circZNF827*-hnRNPK/L mut. pcDNA5-Laccase-*circZNF827*-2XMS2 was created by subcloning a 2xMS2 hairpin sequence (annealed phosphorylated DNA oligos) into pcDNA3.1(+)-Laccase2-*circZNF827* using the unique BspE1 site, *Supplementary file 2c*. Positive clones were verified by sequencing and a HindIII-NotI fragment from this vector was used to insert into the HindIII-NotI sites of pcDNA5-FRT-TO. This construct was used to create stable inducible HEK293-Laccase2-*circZNF827*-2XMS2 cells.

pcDNA5-FLAG-MS2-coat-Twin-Streptag was created by inserting the FLAG-MS2 coat protein open-reading frame (HindIII fragment) from pNMS2-FLAG (*Clement et al., 2011*) into pcDNA5-Twin-Streptag. pcDNA5-Twin-Streptag was made by inserting a PCR product (containing a Twin-Streptag followed by a TEV protease site) using a pDSG-IBA-Twin-Strep-Tag vector as template (IBA-lifesciences) into the HindIII-ApaI site of pcDNA5-FRT-TO (Invitrogen).

To create plasmids for expression of FLAG-tagged RNA binding proteins (RBPs), the coding sequences of RBPs *hnRNPK*, *hnRNPL*, *hnRNPU*, *DDX3X*, and *DHX9* were PCR-amplified from cDNA prepared from RNA isolated from L-AN-5 cells whereas eGFP was sub-cloned from pNEGFP (HindIII/HindIII) and inserted into either KpnI/NotI-, BamHI/NotI- or HindIII-digested pcDNA5_FRT/TO-FLAG. The resulting plasmids were designated pcDNA5_FRT/TO-FLAG-RBP. All plasmids were verified by Sanger sequencing.

## Cell culturing

All used cell lines have been periodically tested for mycoplasma contamination (PCR-based test) with negative results. L-AN-5 cells (obtained from Childrens Oncology Group - COG Cell Line and Xenograft Repository (www.cogcell.org)) were maintained in RPMI whereas SH-SY5Y (obtained from ATCC; ATCC CRL-2266) and HEK293S-Flp-In T-Rex (purchased from Invitrogen) cells were maintained in DMEM medium (Gibco, Dublin, Ireland, 32430100). For all three cell lines, the cell culture medium was supplemented with 10% fetal bovine serum (Gibco, 10082139) and 1% penicillin/streptomycin (Gibco, 15140122). P19 cells (obtained from ATCC; ATCC CRL-1825) were maintained in MEMα supplemented with 7.5% newborn calf serum (Gibco, 26010074), 2.5% fetal bovine serum (Gibco, 10082139), and 1% penicillin/streptomycin (Gibco, 15140122). HEK293 Flp-In T-Rex cells were maintained in DMEM supplemented with 10% tetracyclin-free fetal bovine serum (Gibco, 10082139) and 1% penicillin/streptomycin (Gibco, 15140122). All cells were cultured at 37°C in 5% (v/v) $CO_2$.

For neuronal differentiation of the neuroblastoma cell lines L-AN-5, SH-SY5Y, and P19 10 µM retinoic acid (RA) (Sigma-Aldrich, St. Louis, Missouri, United States) was added to the cell culture medium for 4 days.

The cell line with stable expression of *circZNF827* was generated as previously described (*Hollensen et al., 2018*). Briefly, HEK293 Flp-In T-Rex cells were co-transfected with pcDNA5-FRT/TO-laccase2-*circZNF827*, -*circZNF827*-MS2, or -*circZNF827*-hnRNPK/L mut and a plasmid for expression of the *Flp* recombinase (pOG44). Cell culture medium supplemented with 100 ng/ml Hygromycin (Thermo Scientific, Waltham, Massachusetts, United States) and 10 ng/ml Basticidin S (Thermo Scientific) was used for selection of positive clones. The resulting cell line was designated HEK293 Flp-In T-Rex-*circZNF827*, *circZNF827*-MS2, and -*circZNF827*-hnRNPK/L mut. Tetracycline (Tet) concentrations used for titration of circRNA induction (Northern blot) were 5 ng/ml, 25 ng/ml, 100 ng/ml, and 250 ng/ml, respectively. 25 ng/ml Tet was used for RIP experiments and 250 ng/ml Tet for hnRNP K/L immunofluorescence assays.

## mESC culture and differentiation

E14 mESCs (ES-E14TGs2a; obtained from ATCC; ATCC CRL-1821) were grown on 0.1% gelatin-coated plates in 2i medium (*Ying et al., 2008*) containing: DMEM/F12 (Gibco, 31331) and Neurobasal (Gibco, 12348) 1:1, N2 supplement (Gibco, 17502048), B27 supplement (Gibco, 17504044), 1X glutaMax (Gibco, 35050061), 1X penicilin/streptomycin (Gibco, 15140122), 1 mM sodium pyruvate (Gibco, 11360070), 50 nM 2-mercaptoethanol (Gibco, 31350010), nonessential amino acids (Gibco, 11140076), LIF, 3 µM GSK3 inhibitor (CHIR-99021) and 1 µM MEK inhibitor (PD0325901). They were differentiated into neurons as previously described (*Bibel et al., 2007*) with some modifications. 4 million cells were differentiated into embryoid bodies in suspension in petri dishes for bacterial culture in 15 ml medium containing the same as before, but with 10% FBS and without LIF or GSK3 and MEK inhibitors. Every second day, the medium was changed and the embryoid bodies transferred to fresh petri dishes. On days 4 and 6, 5 µM ATRA (Sigma-Aldrich, R2625) was added to the medium. On day 8 of differentiation, the embryoid bodies were disgregated with 5% trypsin (Gibco, 15400054) and the cells plated in poly-DL-ornithine (Sigma-Aldrich, P8638) and laminin (Sigma-Aldrich, L2020) coated plates in N2 medium, containing DMEM/F12 and neurobasal 1:1, N2 supplement, sodium pyruvate, glutaMax, 15 nM 2-mercaptoethanol, and 50 µg/ml BSA. The medium was changed after 2 hr and after 24 hr. 48 hr after plating the neuronal precursors, the medium was changed to complete medium, containing B27 supplement, in addition to the N2 medium. Neurons were harvested 2 and 8 days after plating. Our E14 cell line has periodically been tested for mycoplasma contamination (PCR-based test) with negative results.

## RNA sequencing and circRNA prediction

20 µg RNA from each sample was depleted of rRNA using a Ribo-Zero rRNA Magnetic Kit (Epicentre, St Louis, Missouri, United States) including the optional RiboGuard RNase inhibitor according to the manufacturer's protocol. The concentration was normalized so that each sample contained the same amount of RNA. To 1/3 of the sample 1/10 of the recommended amount of spike-in (ERCC RNA spike-in mix, Ambion) was added, ethanol precipitated, and resuspended in 'Elute, fragment, finish mix' (Illumina, San Diego, California, United States). The remaining 2/3 of the sample was ethanol precipitated and resuspended in 15 µl nuclease free water. The sample was heated to 70 °C for 1 min and incubated on ice for 2 min. 5 µl RNase R mixture (Epicentre) was added to the sample before incubation at 37°C for 30 min. RNase R was removed by phenol/chloroform extraction. The RNA was resuspended in 'Elute, fragment, finish mix' (Illumina). Sequencing libraries were prepared using Truseq stranded RNA LT kit (Illumina) from both Ribo-Zero and Ribo-Zero/RNase R samples, by fragmentation, 1st and 2nd strand cDNA synthesis, 3'-end adenylation, ligation of adaptors, and enrichment of DNA fragments using the manufacturer's protocol. The quality of the library was validated using an Agilent Bioanalyzer 1000 (Agilent Technologies, Santa Clara, California, United States). The samples were sequenced using the Illumina HiSeq 2500 platform with 100 bp paired-end reads (AROS Applied Biotechnology, Aarhus, Denmark).

Reads were mapped onto the mm10 genome, and circRNAs were detected and quantified using find_circ (*Memczak et al., 2013*), CIRCexplorer2 (*Zhang et al., 2016*) (v2.3.3), and ciri2 (*Gao et al., 2018*) (v2.0.6) using default settings except for find_circ, where a stringent mapq threshold of 40 was used for both adaptor sequences as proposed previously (*Hansen, 2018*). The prediction-output from all pipelines was merged and intersected, and only circRNAs detected by all three pipelines and with three-fold enrichment of backsplice-spanning reads in the RNAseR-treated samples were

defined as bona fide. Expression, based on untreated samples quantified ciri2, was RPM normalized and the top100 expressed bona fide circRNAs across all samples were subjected to kmean clustering using five centers based on within-clusters sum of squared. Annotated genes (UCSC annotation) with at least one splice site in common with circRNAs were denoted as host genes, and based on host-gene annotation, exon numbers and flanking intron lengths were extracted.

The circ-to-linear ratios were based on the backsplice junction spanning reads and the mean of upstream and downstream linear spliced reads as quantified by find_circ.

To compare with human expression profiles, the top100 expressed circRNAs were converted from mm10 to hg19 coordinates using liftOver (UCSC), and only fully matched loci were considered homologous. Data is available at the NCBI Gene Expression Omnibus: GSE157788.

RNAseq from Rybak-Wolf et al (GSE65926 *Rybak-Wolf et al., 2015*) was solely analyzed with find_circ using stringent settings as described above.

## Lentiviral production

Third-generation lentiviral vectors were produced in HEK293T cells as previously described (*Hollensen et al., 2017*). One day before transfection, cells were seeded in 10 cm dishes at a density of $4 \times 10^6$ cells/dish. Transfections were carried out with 3.75 µg pMD.2G, 3 µg pRSV-Rev, 13 µg pMDLg/pRRE and 13 µg lentiviral transfer vector using a standard calcium phosphate or polyethylenimine transfection protocol. Medium was changed to RPMI medium one day after transfection. Two days after transfection, viral supernatants were harvested and filtered through 0.45 µm filters (Sartorius, Göttingen, Germany). All lentiviral preparations were made in at least triplicates and pooled before determination of viral titers.

To determine viral titers of lentiviral preparations, flow cytometric measurements of eGFP expression were used as previously described (*Hollensen et al., 2017*). One day prior to transduction, L-AN-5 cells were seeded at a density of $5 \times 10^5$ cells/well in 12-well plates. For all lentiviral preparations, transductions with $10^2$- and $10^3$-fold dilutions of virus-containing supernatants were carried out. Both viral supernatants and growth medium were supplemented with 4 µg/ml polybrene. One day after transduction, medium was changed. Five days after transduction, cells were harvested and fixated in 4% paraformaldehyde (Sigma-Aldrich). eGFP expression levels were analyzed on a Cyto-FLEX flow cytometer (Beckman Coulter, Brea, California, United States). Lentiviral titers were calculated based on samples with between 1% and 20% eGFP-positive cells using the formula: titer (TU/ml)=F·Cn·DF/V, where F represents the frequency of eGFP-positive cells, Cn the total number of target cells counted the day the transductions were carried out, DF the dilution factor of the virus and V the volume of transducing inoculum.

## circRNA knockdown and differentiation of L-AN-5 cells

One day prior to transduction with lentiviral vectors encoding circRNA-specific dishRNAs, L-AN-5 cells were seeded at a density of $6.6 \times 10^6$ cells/dish in 10 cm dishes, $2.2 \times 10^6$ cells/dish in 6 cm dishes, or $0.8 \times 10^6$ cells/well in 6-well plates. Transductions were carried out using equal MOIs calculated based on titers determined by flow cytometry. Both viral supernatants and growth medium were supplemented with 4 µg/ml polybrene. One day after transduction, medium was changed. Two days after transduction, differentiation was initiated by addition of 10 µM RA (Sigma-Aldrich) to the cell culture medium. The L-AN-5 cells were differentiated for 4 days.

## NGF stimulation

Lentiviral transduction and RA-mediated differentiation of L-AN-5 cells were carried out as described in the section 'circRNA knockdown and differentiation of L-AN-5 cells'. After 4 days of differentiation, the L-AN-5 cells were stimulated with NGF (200 ng/ml) (Thermo Scientific) for 30 min and subsequently harvested for RNA purification.

## mRNA decay assay

Lentiviral transduction and RA-mediated differentiation of L-AN-5 cells were carried out as described in the section 'circRNA knockdown and differentiation of L-AN-5 cells'. The L-AN-5 cells were cultured in 6 cm dishes containing 6 ml cell culture medium supplemented with 10 µM RA. 4 ml cell culture medium was aspirated from each 6 cm dish and pooled from cells transduced with the same

dishRNA. For one dish per dishRNA, the residual medium was aspirated and 3.5 ml of the collected medium was added. For the remaining dishes, the residual medium was aspirated and 3.5 ml of the collected medium supplemented with 2 mM BrU (ThermoFisher) was added. 1 hr after addition of BrU to the cell culture medium, the cells were washed three times in cell culture medium. 50 min after removal of the BrU-containing cell culture medium the first samples including the samples not treated with BrU were harvested. Subsequently, samples were harvested after 3, 6, and 9 hr. Total RNA was purified using 1 ml TRI Reagent (Sigma-Aldrich) according to manufacturer's protocol. circRNA knockdown and differentiation of L-AN-5 cells were verified by RT-qPCR using total RNA as described in the section 'Quantitative PCR'. BrU-labeled RNA was immunoprecipitated as described elsewhere (*Meola et al., 2016*). Briefly, BrU antibodies were conjugated to magnetic beads. 15 µl Dynabeads M-280 Sheep Anti-Mouse IgG (Invitrogen, Carlsbad, California, United States) per sample were washed twice in 1x BrU-IP buffer (20 mM Tris-HCl (pH 7.5)), 250 mM NaCl, 0.5 µg/µl BSA, 20 U/ml RiboLock (Fermentas, Waltham, Massachusetts, United States) and resuspended in 1 ml 1x BrU-IP buffer with heparin (20 mM Tris-HCl (pH 7.5), 250 mM NaCl, 1 mg/ml heparin). After 30 min of incubation at room temperature on a rotator, the beads were washed in 1x BrU-IP buffer. Subsequently, the beads were resuspended in 1 ml 1x BrU-IP buffer supplemented with 0.9 µl mouse BrdU antibody (BD Biosciences, San Jose, California, United States, clone 3D4) per sample and incubated for 1 hr at room temperature on a rotator. The beads were washed three times in 1x BrU-IP buffer and resuspended in 50 µl 1x BrU-IP buffer supplemented with 1 mM 5-BrU per sample. After 30 min of incubation at room temperature on a rotator the beads were washed three times in 1x BrU-IP buffer and resuspended in 50 µl 1x BrU-IP buffer per sample. 25 µg of total RNA was diluted to 200 µl and incubated at 80°C for 2 min. 200 µl 2x BrU-IP buffer with BSA and RiboLock (20 mM Tris-HCl (pH 7.5)), 250 mM NaCl, 1 µg/µl BSA, 80 U/ml RiboLock (Thermo Scientific) and 50 µl beads conjugated with BrdU antibodies were added to the RNA samples. After 1 hr of incubation at room temperature on a rotator, the beads were washed four times in 1x BrU-IP buffer. For elution of immunoprecipitated RNA, the beads were resuspended in 200 µl 0.1% SDS. RNA was purified by phenol/chloroform extraction, ethanol precipitation and the RNA pellets were resuspended in 10 µl nuclease free water. 2 µl of immunoprecipitated RNA was used for quantification of mRNA expression levels by RT-qPCR as described in the section 'Quantitative PCR' except that DNase treatment was omitted and 1 µg yeast RNA (Roche, Basel, Switzerland) was added in the cDNA reaction.

## BrU-labeling and immunoprecipitation of newly synthesized RNA

The BrU-labeling and immunoprecipitation of newly labeled RNA were carried out as for the mRNA decay assay except that the cells were harvested 45 min after addition of BrU to the cell culture medium. Furthermore, after binding of the RNA to the beads, the beads were washed once in 1x BrU-IP buffer, twice in 1x BrU-IP buffer supplemented with 0.01% Triton X-100 and twice in 1x BrU-IP buffer.

## Subcellular fractionation of nuclear and cytoplasmic RNA

Subcellular fractionation of nuclear and cytoplasmic RNA was carried out as previously described (*Hollensen et al., 2018*). Briefly, cells were washed in PBS, then 800 µl PBS was added and the cells were scraped off. 100 µl of the cell solution was centrifuged at 12,000 rpm for 10 s at 4°C. Cell pellets were used for purification of total RNA using 1 ml of TRI Reagent (Sigma-Aldrich) according to manufacturer's protocol. The remaining 700 µl of the cell solution was used for subcellular fractionation of nuclear and cytoplasmic RNA. After centrifugation at 12,000 rpm for 10 s at 4°C 300 µl lysis buffer (20 mM Tris-HCl (pH 7.5), 140 mM NaCl, 1 mM EDTA, 0.5% Igepal-630 (Nonidet P-40)) were added to the cell pellets, which were then incubated on ice for 2 min and centrifuged at 1000 g for 4 min at 4°C. Cytoplasmic RNA was purified from the supernatants using 1 ml TRI Reagent (Sigma-Aldrich) according to the manufacturer's protocol. Pellets were washed twice in 500 µl lysis buffer, subjected to a single 5 s pulse of sonication at the lowest settings (Branson Sonifier 250) and nuclear RNA was purified using 1 ml TRI Reagent (Sigma-Aldrich) according to the manufacturer's protocol.

## Subcellular fractionation of nuclear and cytoplasmic protein

Cells were washed in PBS, then 800 µl PBS were added and the cells were scraped off. 80 µl of the cell solution was centrifuged at 500 g for 5 min and 200 µl lysis buffer (1x TBS, 0.5% Igepal-630 (Nonidet P-40)) were added to the cell pellets for isolation of total protein. The remaining 720 µl of the cell solution was used for subcellular fractionation of nuclear and cytoplasmic protein. After centrifugation at 12,000 rpm for 10 s at 4°C cell 300 µl lysis buffer were added and the cell pellets, which were incubated on ice for 2 min and centrifuged at 1000 g for 4 min at 4°C. The supernatants (cytoplasmic fractions) were transferred to new tubes. Pellets (nuclear fractions) were washed twice in 500 µl lysis buffer and once in 500 µl 1x TBS and resuspended in 200 µl lysis buffer. All samples were subjected to two 5 s pulses of sonication at the lowest settings (Branson Sonifier 250) followed by centrifugation at 4000 g for 25 min at 4°C. Supernatants were transferred to new tubes containing 87% glycerol (final concentration of 10%) and concentrations were adjusted using Bio-Rad protein assay (Bio-Rad, Hercules, California, United States).

## Quantitative PCR

RNA was purified using TRI reagent (Thermo Scientific) according the to manufacturer's protocol. RNA samples were treated with DNase I (Thermo Scientific) according to the manufacturer's protocol. First-strand cDNA synthesis was carried out using the Maxima First Strand cDNA synthesis Kit for qPCR (Thermo Scientific) according to the manufacturer's protocol. qPCR reactions were prepared using gene-specific primers (*Supplementary file 2d*) and Platinum SYBR Green qPCR Supermix-UDG (Thermo Scientific) according to the manufacturer's protocol. An AriaMx Real-time PCR System (Agilent Technologies) was used for quantification of RNA levels and the $X_0$ method was used for calculations of relative RNA levels (*Thomsen et al., 2010*) normalized to either GAPDH or beta-actin (ACTB) mRNA as indicated.

## NanoString

Gene expression analysis of 770 neuropathology-related genes were analyzed using the nCounter Human Neuropathology Panel (NanoString Technologies, Seattle, Washington, United States) and the nCounter *SPRINT* Profiler (NanoString Technologies) according to manufacturer's protocol. Data analysis was carried out in the nSolver 4.0 software (NanoString Technologies) using the default settings in the nCounter Advanced Analysis Software (NanoString Technologies).

## Cell cycle assay

Lentiviral transduction and RA-mediated differentiation of L-AN-5 cells were carried out as described in the section 'circRNA knockdown and differentiation of L-AN-5 cells'. Labeling of newly synthesized DNA was carried out using Click-iT Plus EdU Alexa Flour 647 Flow Cytometry Assay Kit (Thermo Scientific) according to manufacturer's protocol. Notably, the cell culture medium of L-AN-5 cells cultured in six-well plates was supplemented with 10 µM EdU for 1.5 hr. To stain total DNA, cells with already detected EdU were resuspended in 400 µl 1x Click-iT saponin-based permeabilization and wash reagent from the Click-iT Plus EdU Alexa Flour 647 Flow Cytometry Assay Kit (Thermo Scientific). Subsequently, RNase A was added to a final concentration of 0.2 mg/ml. After 5 min of incubation at room temperature, propidium iodide was added to a final concentration of 5 µg/ml and the cells were incubated for 30 min at room temperature. Incorporated EdU and total DNA levels were analyzed on a BD LSRFortessa flow cytometer (BD Biosciences). Data analysis was carried out in the FLOWJO software (BD Biosciences). The gating strategy is shown in *Figure 2—figure supplement 3A*.

## Western blotting

Cells were scraped off, pelleted and lysed for 15 min on ice in RSB100 (10 mM Tris-HCl (pH 7.4), 100 mM NaCl, 2.5 mM MgCl2) supplemented with 0.5% Triton X-100 and 1 pill Complete protease inhibitor cocktail (Roche). The cell lysates were subjected to two 5 s pulses of sonication at the lowest settings (Branson Sonifier 250) followed by centrifugation at 4000 g for 15 min at 4°C. Glycerol was added to the supernatants (final concentration: 10%) and protein concentrations were adjusted using Bio-Rad protein assay (Bio-Rad). The protein samples were diluted in 6x loading buffer (9.8% glycerol, 12% SDS, 375 mM Tris-HCl (pH 6.8), 0.03% bromophenol blue, 10% β-mercaptoethanol),

heated at 95°C for 3 min and seprated on a Novex WedgeWell 4–12% Tris-Glycine Gel (Invitrogen). Proteins were transferred to an PVDF Transfer Membrane (Thermo Scientific) using standard procedures. The membranes were blocked in 5% skimmed milk powder in PBS for 1 hr at room temperature. The membranes were incubated at 4°C overnight with primary antibodies diluted as indicated in *Supplementary file 2f* in 5% skimmed milk powder in PBS. After three times wash, the membranes were incubated with goat polyclonal HRP-conjugated secondary antibodies (Dako, Glostrup, Denmark) diluted 1:20,000 in 5% skimmed milk powder in PBS. After 1 hr of incubation at room temperature, the membranes were washed three times and the bound antibodies were detected using the SuperSignal West Femto maximum sensitivity substrate (Thermo Scientific) according to the manufacturer's protocol and using the LI-COR Odyssey system (LI-COR Biosciences, Lincoln, Nebraska, United States).

## In vitro transcription

As DNA templates for in vitro transcription, pcDNA3/circRNA vectors encoding the full-length exonic sequences of five human circRNAs were used. Biotinylated RNAs were produced from 0.5 µg linearized, and phenol/chloroform extracted template using the MEGAscript T7 Transcription Kit (Ambion, Austin, Texas, United States), according to the manufacturer's protocol with addition of 0.75 mM Biotin-14-CTP (Invitrogen) to the transcription reaction. In controls, nuclease free water was added instead of Biotin-14-CTP. The transcribed RNA was purified by phenol/chloroform extraction and dissolved in nuclease free water.

## Streptavidin-biotin pull-down

For each pull-down, 125 µL (bead volume) Pierce Streptavidin magnetic beads (Thermo Scientific) pre-washed in NET-2 buffer (50 mM Tris-HCl pH 7.5, 150 mM NaCl, 0.1% Triton X-100) were incubated with 30 µg in vitro synthesized circRNAs or 30 µg control RNA in 500 µL NET-2 buffer for 1 hr. 4°C mixing end-over-end. The conjugated beads were washed once in NET-2 buffer and incubated with 1.5 mL cell lysate prepared as follows: For each pull down, one 90% confluent 150 mm plate of differentiated L-AN-5 cells was washed in 10 mL ice cold PBS, and subjected to cell lysis in 1.5 mL hypotonic gentle lysis buffer (10 mM Tris-HCl pH 7.5, 10 mM NaCl, 2 mM EDTA pH 8.0, 0.1% Triton X-100) supplemented with Complete protease inhibitor cocktail (Roche, 1 pastel per 10 mL lysis buffer) for 5 min. on ice. Cells were collected by scraping and re-suspension, then supplemented with NaCl to 150 mM final concentration and incubated on ice for 5 min. Cleared cell lysate was obtained by centrifugation at 14,000 rpm, 4°C, 10 min. and supplemented with 10 µL Ribolock RNase Inhibitor (40 U/µL, Thermo Scientific) per 10 mL lysis before incubation with circRNA-coupled streptavidin beads for 1.5 hr, 4°C mixing end-over-end. From the cleared lysate 1% was mixed 1:1 with 2xSDS-load buffer (20% glycerol, 4% SDS, 100 mM Tris-HCL pH 6.8, 0.05% Bromophenol blue/ Xylene cyanol and 10% β-mercaptoethanol) and kept as input sample. Following capture of proteins, beads were washed four times in NET-2 buffer and bound proteins were eluted in 40 µL preheated 2x SDS-load buffer by boiling at 90°C for 5 min. Eluates were subjected to SDS-PAGE electrophoresis and run either completely through and stained with SilverQuest Silver staining kit (Life Technologies, Carlsbad, California, United States) according to the manufacturer's protocol, or only 1.5 cm into the gel for subsequent staining with GelCode Blue Stain Reagent (Thermo Scientific) according to the manufacturer's protocol and excision of the bands for mass spectrometry application (see below).

## Protein analysis by nano-LC-MS/MS

Interacting proteins were identified and quantified according to previously described methods (*Britze et al., 2014*). Briefly, each gel lane was cut into 1 × 1 mm pieces and cysteine residues were blocked by reduction and alkylation using tris(2-carboxyethyl)phosphine and iodoacetamide, respectively. In-gel digestion was performed using trypsin and resulting peptides were extracted from gel pieces using acetonitrile and trifluoroacetic acid and finally purified on PepClean C-18 Spin columns (Thermo Scientific). Liquid chromatography tandem mass spectrometry (LC-MS/MS) was performed on an EASY nanoLC coupled to a Q Exactive Plus Hybrid Quadrupole-Orbitrap Mass Spectrometer (Thermo Scientific). Peptide samples were separated on a C-18 reverse phase column (EASY-Spray PepMap from Thermo Scientific with 25 cm length, 75 µm inner diameter, and 2 µm particle size)

and eluted by a 90 min linear gradient of acetonitrile (4–40%) containing 0.1% formic acid. The MS was operated in data dependent mode, automatically switching between MS and MS2 acquisition, with mass resolution of 70,000 and 17,500, respectively. Up to 10 most intense ions were fragmented per every full MS scan, by higher energy collisional dissociation. Dynamic exclusion of 10 s was applied and ions with single charge or unassigned charge states were excluded from fragmentation.

MaxQuant software version 1.5.2.8 was applied for protein identification and label-free quantification by means of peptide peak areas (*Cox and Mann, 2008*). MS raw files were searched against a database consisting of 20,197 *Homo sapiens* sequences downloaded from Uniprot.org, August 2015. Carbamidomethylation of cysteines was set as a fixed modification, whereas methionine oxidation and protein N-terminal acetylation were set as dynamic modifications. The false discovery rate (FDR) was assessed by searching against a reverse decoy database, and FDR thresholds of protein and peptide identification were both set to 0.01.

## Immunofluorescence

For indirect immunofluorescence experiments, $1 \times 10^5$ HEK293 Flp-In T-Rex or *circZNF827*_HEK293 Flp-In T-Rex cells were grown directly on poly-L-lysine coated coverslips in 12-well plates. Transcription of circRNA transgene was induced by addition of 10–250 ng/ml tetracycline and the induction profile was tested by Northern blotting in a parallel experiment. 24 hr later cells were fixed in 4% paraformaldehyde for 15 min, and permeabilized and blocked with PBS/1% goat serum (or horse serum)/0.5% Triton X-100 for 20 min. Cells were then incubated for 1–16 hr with mouse anti-hnRNPK (Abcam, Cambridge, United Kingdom), Rabbit anti-hnRNPU (Santa Cruz Biotechnologies, Dallas, Texas, United States) or mouse anti-hnRNPL (Abcam). Antibodies were used at 1:1000 dilutions. Following removal of the primary antibody, cells were incubated for 1 hr with 4 µg/mL secondary anti-IgG antibodies labeled with Alexa-594 and Alexa-488 (Molecular Probes, Eugene, Oregon, United States).

## RNA immunoprecipitation and co-immunoprecipitation of proteins

For RNA immunoprecipitation (IP) and co-immunoprecipitation (co-IP) of proteins L-AN-5 cells were seeded at a density of $6.6 \times 10^6$ cells/dish in 10 cm dishes and differentiated as described in the section 'Cell culturing'. HEK293 Flp-In T-Rex cells and HEK293 Flp-In T-Rex *circZNF827* cells were seeded a density of $6.6 \times 10^6$ cells/dish in 10 cm dishes. HEK293 Flp-In T-Rex and HEK293 Flp-In T-Rex *circZNF827* cells were transfected with 5 µg pcDNA5/FRT-TO-FLAG-RBP and 25 µl polyethylenimine (PEI) (1 µg/µl) according to a standard PEI transfection protocol. 6 hr after transfection, RBP and circRNA expression were induced by addition of 100 ng/ml tetracycline to the cell culture medium. For IP of endogenously expressed proteins, antibodies were conjugated to Protein G dynabeads (Thermo Scientific) prior to harvest of the cells. 25 µl beads per sample were washed three times in 1 ml NET-2 buffer (50 mM Tris-HCl pH 7.5, 100 mM NaCl, 0.1% Triton-X100). Subsequently, the beads were resuspended in 800 µl NET-2 buffer per sample and added 10 µl hnRNP K, hnRNP L, or IgG antibody per sample. After conjugation for 120 min at 4°C on a rotator, the beads were washed twice in NET-2 buffer and resuspended in 50 µl NET-2 buffer per sample. For IP of FLAG-tagged proteins, 50 µl anti-FLAG-M2 agarose slurry was washed twice in 1.5 ml NET-2 buffer and resuspended in 50 µl NET-2 buffer per sample. The cells were lysed after a single wash in PBS by addition of 1 ml ice-cold hypotonic gentle lysis buffer (10 mM Tris-HCl pH 7.5, 10 mM NaCl, 1 mM EDTA, 0.25% Triton-X100, and one pill Complete protease inhibitor cocktail (Roche) per 10 ml), scraped off, and transferred to an Eppendorf tube. After incubation for 5 min on ice, 35 µl 4 M NaCl (final 150 mM) was added and the samples were incubated for 2 min on ice. The lysates were subjected to a single 5 s pulse of sonication at the lowest settings (Branson Sonifier 250) and centrifuged at 13,000 rpm for 15 min at 4°C. For input protein and RNA controls, 50 µl and 100 µl of the lysate were resuspended in 50 µl 2xSDS-load buffer (20% glycerol, 4% SDS, 100 mM Tris-HCL pH 6.8, 0.05% Bromophenol blue/Xylene cyanol and 10% β-mercaptoethanol) and 1 ml TRI Reagent (Sigma-Aldrich), respectively. The input protein control samples were incubated for 3 min at 80–90°C before storage at −20°C. The remainder of the supernatants was transferred to tubes containing 50 µl bead slurry and nutated at 4°C for 2 hr. Subsequently, the beads were washed seven times in 1.5 ml ice-cold NET-2 (5 min per wash) and protein was eluted form one third of the beads by addition of 100

µl 2xSDS-load buffer followed by incubation for 3 min at 80–90℃, whereas RNA was eluted from two thirds of the beads by addition of 1 ml TRI Reagent (Sigma-Aldrich).

## Northern blotting

Northern blots were carried out as previously described in *Damgaard and Lykke-Andersen, 2011*. Briefly, 10 µg RNA was separated in a 1.2% formaldehyde-agarose gel. Subsequently, the RNA was transferred to a Hybond membrane (GE Healthcare, Chicago, Illinoise, United States). The membrane was hybridized with *circZNF827*- or *ACTB*-specific [32P]-end-labeled oligonucleotides (sequences are specified in *Supplementary file 2e*) overnight and subsequently exposed on phosphorimager screens and visualized on a Typhoon FLA 9500 (GE Healthcare).

## ChIP

Lentiviral transduction and RA-mediated differentiation of L-AN-5 cells were carried out as described in the section 'circRNA knockdown and differentiation of L-AN-5 cells'. HEK293 Flp-In T-Rex, HEK293 Flp-In T-Rex *circZNF827*, and HEK293 Flp-In T-Rex *circZNF827*-MS2 cells were seeded at a density of $5 \times 10^6$ cells/dish in 10 cm dishes. Transfections were carried out with 5 µg pcDNA5-FLAG-MS2-coat-Twin-Streptag encoding the MS2 coat protein and 25 µl PEI (1 µg/µl) according to a standard PEI transfection protocol. 5 hr after transfection, MS2 coat protein and circRNA expression were induced by addition of 100 ng/ml tetracyclin to the cell culture medium. The ChIP assay including crosslinking and harvest of cells were carried out using the Pierce Magnetic ChIP Kit (Thermo Scientific) according to the manufacturer's protocol except that sonication was carried out on a Covaris S2 ultrasonicator (settings: burst: 15%, cycles: 200, intensity: 6, cycle time: 20 min, frequency sweeping: on, de-gas: on) and 10 µl Anti-FLAG M2 beads (Sigma-Aldrich) blocked with 10 ng/ml FLAG-peptid (Sigma-Aldrich) were used for IP of the FLAG-tagged MS2 coat protein. The antibodies used for the ChIP assay are listed in *Supplementary file 2f*. DNA fragments were quantified as described in the section 'Quantitative PCR' using the gene-specific primers listed in *Supplementary file 2d*.

## Statistical analysis

In biochemical assays (conducted in at least biological triplicates), the significance of difference between samples were calculated by a two-tailed Student's t test to test the null hypothesis of no difference between the two compared groups. The assumption of equal variances was tested by an F test. $p < 0.05$ was considered statistically significant. Data are presented as mean ± SD.

# Acknowledgements

Proliferation assays using flow cytometry was performed at the FACS Core Facility, Aarhus University, Denmark. Karina Hjorth and Maria Vad Jakobsen are thanked for excellent technical assistance. Thanks to Serafin Pinol-Roma for sharing anti-hnRNP C1/C2 antibody.

# Additional information

### Competing interests

Torben Heick Jensen: Reviewing editor, *eLife*. The other authors declare that no competing interests exist.

### Funding

| Funder | Grant reference number | Author |
| --- | --- | --- |
| Novo Nordisk | NNF17OC0028804 | Anne Kruse Hollensen |
| Novo Nordisk | NNF15OC0017598 | Anne Kruse Hollensen |
| Lundbeckfonden | R140-2013-13425 | Anne Kruse Hollensen |
| Lundbeckfonden | R289-2018-1504 | Christian Kroun Damgaard |
| Danish Cancer Society | R167-A11105 | Andreas Bjerregaard Kamstrup |

| Aarhus Universitets For-skningsfond | NOVA grant | Henriette Sylvain Thomsen |

The funders had no role in study design, data collection and interpretation, or the decision to submit the work for publication.

## Author contributions

Anne Kruse Hollensen, Conceptualization, Data curation, Formal analysis, Investigation, Methodology, Writing - review and editing; Henriette Sylvain Thomsen, Data curation, Formal analysis, Investigation, Methodology, Writing - review and editing; Marta Lloret-Llinares, Formal analysis, Investigation, Methodology, Writing - review and editing; Andreas Bjerregaard Kamstrup, Formal analysis, Investigation, Writing - review and editing; Jacob Malte Jensen, Data curation, Formal analysis, Investigation; Majbritt Luckmann, Investigation, Methodology; Nanna Birkmose, Investigation; Johan Palmfeldt, Data curation, Formal analysis, Supervision, Investigation, Methodology, Writing - review and editing; Torben Heick Jensen, Supervision, Investigation, Methodology, Writing - review and editing; Thomas B Hansen, Resources, Data curation, Formal analysis, Validation, Investigation, Methodology, Writing - review and editing; Christian Kroun Damgaard, Conceptualization, Data curation, Formal analysis, Supervision, Funding acquisition, Investigation, Writing - original draft, Project administration, Writing - review and editing

## Author ORCIDs

Anne Kruse Hollensen https://orcid.org/0000-0002-5461-6893
Marta Lloret-Llinares http://orcid.org/0000-0002-8205-9799
Torben Heick Jensen http://orcid.org/0000-0001-5127-1239
Thomas B Hansen https://orcid.org/0000-0002-7573-9657
Christian Kroun Damgaard https://orcid.org/0000-0003-4940-0868

## Decision letter and Author response

Decision letter https://doi.org/10.7554/eLife.58478.sa1
Author response https://doi.org/10.7554/eLife.58478.sa2

# Additional files

## Supplementary files

• Supplementary file 1. Table of high-confidence circRNAs identified in the stages mESC, NPC and N8 neurons. Genomic coordinates and overlapping gene symbols are given along with expression level (RPM) at the indicated stage.

• Supplementary file 2. Lists of candidate circRNAs, differentially regulated genes upon circZNF827 knockdown, oligonucleotides and antibodies used in the study. (a) List of circRNA candidates selected for knockdown in L-AN-5 cells and their respective mouse homologues. (b) List of differentially regulated genes (>2 fold change, p<0.05) assessed by nanoString hybridization (+/- BrU labeling), when comparing control (Irr) vs. circZNF827 knockdown (+/- RA). (c-e) List of oligonucleotides used in the study. (f) List of antibodies used in the study.

• Transparent reporting form

## Data availability

RNA sequencing data related to Figure 1 and Figure 1 supplement figure 1 has been deposited in GEO under the accesion code: GSE157788.

The following dataset was generated:

| Author(s) | Year | Dataset title | Dataset URL | Database and Identifier |
|---|---|---|---|---|
| Hollensen AK, Thomsen HS, Lloret-Llinares M, | 2020 | circZNF827 nucleates a transcription inhibitory complex to balance neuronal differentiation | https://www.ncbi.nlm.nih.gov/geo/query/acc.cgi?acc=GSE157788 | NCBI Gene Expression Omnibus, GSE157788 |

Kamstrup AB, Jensen JM, Luckmann M, Birkmose N, Palmfeldt J, Jensen TH, Hansen TB, Damgaard CK

The following previously published datasets were used:

| Author(s) | Year | Dataset title | Dataset URL | Database and Identifier |
|---|---|---|---|---|
| Rybak-Wolf A, Stottmeister C, Glazar P, Pino N, Giusti S, Hanan M, Behm M, Bartok O, Ashwal-Fluss R, Jens M, Herzog M, Schreyer L, Papavasileiou P, Ivanov A, Öhman M, Refojo D, Kadener S, Rajewsky N | 2015 | Circular RNAs in the mammalian brain are highly abundant, conserved, and dynamically expressed | https://www.ncbi.nlm.nih.gov/geo/query/acc.cgi?acc=GSE65926 | NCBI Gene Expression Omnibus, GSE65926 |

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
