## [Decision Letter]

**Acceptance summary:**

Circular RNAs have been previously shown to be noncoding RNAs capable of titrating away microRNAs or RNA binding proteins. Here the authors show a role for a circular RNA at chromatin for control of gene expression. circZNF877 is shown to interact with its host zinc finger protein and two RNA binding proteins to nucleate a gene repressor complex at target gene loci.

**Decision letter after peer review:**

Thank you for submitting your article "*circZNF827* nucleates a transcription inhibitory complex to balance neuronal differentiation" for consideration by *eLife*. Your article has been reviewed by two peer reviewers, and the evaluation has been overseen by a Reviewing Editor and James Manley as the Senior Editor. The reviewers have opted to remain anonymous.

The reviewers have discussed the reviews with one another and the Reviewing Editor has drafted this decision to help you prepare a revised submission.

Summary:

Hollensen and coworkers identified the circRNA at different various neuronal differentiation stages using three different circRNA prediction tools and found circZFP827 together with other circRNAs have temporal expression patterns during differentiation. The authors decided to focus on circZFP827 and showed that downregulation of *circZNF827* increases the expression of several neuronal genes and reduces the S-phase fraction. Later, the authors found that circ-FP827 controls the expression of two retinoid receptors (RARa and RARg) posttranscriptionally. Using Nanostring, the authors showed that *circZNF827* knockdown increases the expression of several neuronal genes (*NGFR*, etc.) and subsequently a known downstream gene of *NGFR*, c-fos. To understand how *circZNF827* regulates the expression of downstream genes transcriptionally or post-transcriptionally, the authors performed pull-down followed by LC-MS/MS and identified hnRNP K and L as its main interactors. Next, the authors found out that knocking down of circZFP827 increases the cytoplasmic level of hnRNP K and overexpressing of circZFP827, in contrast, generates multiple distinct nuclear foci. Finally, the authors showed that circZNF287 downregulates *NGFR* transcription by interacting with hnRNP k inhibitory complex and the ZNF827 protein.

While the reviewers find the paper to be of interest, there are several points that require clarification with additional data before the paper can be considered for publication. The key points relate to the copy number of nuclear *circZNF827* and its mechanism of action through affecting nuclear RNA binding proteins hnRNPK and hnRNPL.

Essential revisions:

1) The authors' model is that the small (10%) amount of the circRNA that is present in the nucleus has a functional role via interacting with hnRNP K and L. This is possible but more direct functional experiments should be included to justify the model. For example, the authors should express dishRNA-resistant versions of *circZNF827* and show that the wildtype circRNA rescues the *NGFR* expression phenotypes but a circRNA with mutated K and L binding sites does not. The authors should also use a technique like ChIRP to show direct binding of the circRNA to the *NGFR* locus.

2) The retinoic acid receptor data (Figure 3) is poorly integrated into the rest of the manuscript and the data quality is weak. The supplemental figure shows that RARalpha has a slight (20%) increase in transcription that is insufficient to explain the mRNA effect sizes in the main figure. There is no significant change in RARgamma transcription or decay rates, yet the mRNA expression is increased. Most critically, the authors have failed to connect the proposed hnRNP K/L based mechanism (Figures 5-7) to RAR transcriptional regulation.

3) The biochemical data supporting a complex comprising hnRNP K and L is weak and confusing. hnRNP K can pull down a little bit of L, but hnRNP L pull down does not appear to pull down K (Figure 5E, 6D). This is relevant to the author's main model as the circRNA can efficiently bind L, but hnRNP K is the one that seems to be having the transcriptional control role.

4) Some circRNAs in group 4 (Figure 1E) do not have the same expression pattern upon validation in qPCR (Figure 1G) (see circEZH2, circZFP827). If *circZNF827* represses neuronal genes and knocking down *circZNF827* upregulates pan neuronal genes (*Tubb3*, *Map2*, *Nefl* and *TrkB*, Figure 2B), it is counterintuitive to see the circ expression level upregulated during differentiation by RT-qPCR (Figure 1G, the color scale should be replaced to allow proper visualization)?

5) The authors showed that there are putative hnRNP-L or K binding sites in *circZNF827* (Figure 5—figure supplement 1). Would the deletions or mutations of hnRNP-L or K binding sites in *circZNF827* disrupt its interactions with hnRNP-L or K and later the *circZNF827*-hnRNPK/L-ZNF827 binding to the promoter of *NGFR*?

---

## [Author Response]

Essential revisions:1) The authors' model is that the small (10%) amount of the circRNA that is present in the nucleus has a functional role via interacting with hnRNP K and L. This is possible but more direct functional experiments should be included to justify the model. For example, the authors should express dishRNA-resistant versions of circZNF827 and show that the wildtype circRNA rescues the NGFR expression phenotypes but a circRNA with mutated K and L binding sites does not.

We agree that a rescue experiment, if successful, would be an important addition. Our model is based on the finding that *circZNF827* is regulating *NGFR* at the transcriptional level, which could potentially be an indirect effect, although hnRNPK (and now also ZNF827) binding to the *NGFR* promoter is decreased upon *circZNF827* knockdown, and although we observe nucleation/phase separation of endogenous hnRNPK and -L upon overexpression of *circZNF827*.

Previously, we had already conducted experiments with overexpression of *circZNF827* (on top of the endogenous level) from a lentiviral Laccase2-driven circRNA expression system in differentiated L-AN-5 cells. Curiously, this did not alter expression from *NGFR*, *NEFL*, *TrkB*, *MAP2* or *TUBB3*, suggesting either of three scenarios (not necessarily mutually exclusive): 1) that the level of *circZNF827* is already saturated, 2) that high expression will lead to non-physiological titration effects, or 3) that the backsplicing junction found only in the endogenous circRNA is important for its structure, regulation and function.

Rescue experiments are inherently difficult to conduct for circRNAs compared to protein-coding mRNAs, for 2 main reasons: 1) It is difficult to control exogenous circRNA expression levels due to its ultra-high stability, and 2) Since efficient circRNA expression vectors that generate high-levels of circRNA (relative to undesired linear precursor RNA or concatemer RNAs) often have predesigned splice sites (e.g. the Laccase2 system), which will produce a circRNA with a different backsplicing junction, this structure may be important for the function of the endogenous circRNA. Thus, with current circRNA expression vector technologies, there is a *case-by-case* trade-off between efficiently producing a significantly different circRNA with potentially altered structure and function and a more subtle mutant with low circularization efficiency.

With these limitations in mind, and to directly address the reviewer’s point, we conducted a series of knockdowns using dishRNAs and attempted to rescue upregulation of *NGFR* by expressing two newly cloned variants of dishRNA-resistent (changed backsplicing junction) circRNAs from a laccase2-driven lentiviral vector: Full-length *circZNF827* or a hnRNPK/L binding site deletion mutant (very poorly bound by hnRNPK/L in RNA-immunoprecipitation experiments – see below in comments to point 5). High expression (using 3 different MOI’s) of the full-length laccase2-driven *circZNF827*, left *NGFR* expression unaltered, at best only marginally lowered as judged by western blotting analysis (no significant change in biological triplicates). Similarly, at the level of RT-qPCR, we did not observe a normalization of the upregulation of *NGFR* expression. However, the hnRNP K/-L binding site mutant circRNA was, as would be expected, unable to change *NGFR* expression levels back to the normal.

As stated above, we conclude that there are likely sequences, or structures, surrounding the backsplicing junction, that are important for the function of the produced dishRNA-resistant circRNA. Also, expression levels might still be too high and difficult to decrease without leaving a large proportion of cells untransduced, which is not a feasible option, when re-introducing expression and assaying biochemically from bulk culture. Additionally, we predict that the timing of re-expression relative to knockdown could also be important not to exceed a “point of commitment” of *NGFR* upregulation, leaving additional time-consuming optimizations of MOI’s and timings of superinfections very unfeasible.

Still, to further solidify our model, we have now performed knockdown of both *circZNF827* and ZNF827 itself, since the circRNA host-encoded protein is an integral part of the circRNA-hnRNPK/L complex. In support of our model, the double knockdown further increases the output from *NGFR* significantly, as was also the case for hnRNPK/L and *circZNF827* co-depletions. In addition, we have now performed biological triplicate ChIP assays using both hnRNP K and ZNF827 antibodies to show that these factors are indeed recruited to the *NGFR* promoter, and that this recruitment is significantly lowered upon *circZNF827* knockdown. We believe that these new results, and further results below, support our model and strengthen our conclusions considerably. The results have been integrated into Figure 6C-D, F-H and Figure 6—figure supplement 1D.

The authors should also use a technique like ChIRP to show direct binding of the circRNA to the NGFR locus.

We agree that the evidence for a direct (or hnRNPK/L/ZNF827 complex-mediated) interaction of *circZNF827* with the *NGFR* promoter (and potentially other promoters) is circumstantial and could be strengthened. In order to address this issue, we find the suggested ChIRP-DNA strategy unsuitable, in this particular case, for two main reasons: First, the circRNA of interest is only 402 nt long and is predicted to be bound along almost the entire one side of its suggested structure by hnRNPK/L at presumably single-stranded and accessible regions – this is now confirmed by our new RIP analyses using the hnRNPK/L binding site mutant of *circZNF827* (see below in comments to point 5). This leaves very little room for design of a suitable number of appropriate tiling probes and control probes (probes that have been shifted to control for direct probe binding and off-target effects). We have previously performed ChIRP to capture another short circRNA for subsequent mass spectrometry analyses without positive results. In comparison, the Chang lab, who carefully developed the procedure, used 48 probes to map genomic interactions sites for HOTAIR lncRNA, which is 2200 nt in length (Chu et al., 2011). In addition, 43 probes were used to the interacting proteome for the 17000 nt long XIST lncRNA and a prominent ChIRP-MS study conducted on circRNA targets (Chen et al., 2017), used 8 and 15 probes, respectively, to pull-down two exogenously expressed circRNAs of approximately 1500 nt each. We therefore predict a poor pull-down efficiency given the limited ability of designing a suitable number of probes. Second, another caveat using this strategy, is that any probes designed to target the short *circZNF827*, will also target its linear host pre-mRNA and its fully processed mRNA, potentially giving rise to false positive signals or out-titration of the probes. Thus, we anticipate that ChIRP will be inefficient and take much optimization and therefore remains unfeasible.

As alternative approaches, we conducted two independent experiments: First, we cloned a 2 x MS2 hairpintagged version of *circZNF827*, which retains its putative hnRNPK/L binding sites (Figure 7A). We established inducible stable HEK293S Flp-In T-Rex cell lines expressing these constructs. Using these cells for transfections with FLAG-MS2 coat protein expression constructs, we performed anti-FLAG ChIP RT-qPCR with amplicons for the *NGFR* promoter region. By this setup, we observed an enriched *NGFR* promoter signal for the MS2tagged *circZNF827* cell line as compared to non-MS2 tagged or non-circRNA controls, while the low control GAPDH promoter-signal remained constant in all cases (Figure 7B). We conclude that the MS2-tagged circZNF287 is also localized to the *NGFR* promoter region in accordance with our model. Importantly, hnRNPK and -L both interact with MS2-tagged circRNA in RNA immunoprecipitation assays (similar to *circZNF827* WT affinity) (Figure 7—figure supplement 1). Second, since ZNF827 interacts with hnRNPK/L and *circZNF827*, we performed ChIP of endogenous ZNF827 in L-AN-5 cells to address whether the host protein also interacts with the *NGFR* promoter. Indeed, our results showed an enrichment of ZNF827 at the promoter when *circZNF827* is present in cells, while its knockdown significantly diminishes the occupancy of ZNF827 at the promoter (Figure 6D). This places hnRNPK, ZNF827 and importantly, *circZNF827* at the *NGFR* promoter, consistent with a likely nucleation-mechanism and repression of transcription. Most importantly, the recruitment of both ZNF827 and hnRNP K/L to the *NGFR* promoter is dependent on *circZNF827*. These results have considerably strengthened our model and have been integrated into Figure 6 and 7, along with a discussion in the main text as indicated.

2) The retinoic acid receptor data (Figure 3) is poorly integrated into the rest of the manuscript and the data quality is weak. The supplemental figure shows that RARalpha has a slight (20%) increase in transcription that is insufficient to explain the mRNA effect sizes in the main figure. There is no significant change in RARgamma transcription or decay rates, yet the mRNA expression is increased. Most critically, the authors have failed to connect the proposed hnRNP K/L based mechanism (Figures 5-7) to RAR transcriptional regulation.

We agree that the effects on Retinoic Acid Receptor α (RARa) and γ (RARg) expression are only moderate and in the latter case not significant in terms of BrU pulse-measured transcription. To strengthen these data and to integrate them better into the hnRNP K/L-ZNF827 regulatory model as suggested, we have performed ChIP-qPCR of hnRNPK and ZNF827 on the promoter of the RARa gene (Figure 6—figure supplement 1D). Importantly, and as observed with the *NGFR* promoter, we detect less docking of hnRNPK and ZNF827 during *circZNF827* knockdown, indicating that the hnRNPK/L-regulatory model is likely not limited to the *NGFR* locus. The reason for the small, yet significant, increase in BrU incorporation (RARa) upon *circZNF827* knockdown (compared to steady-state level) might be a temporal issue, when comparing steadystate levels (5-6 days in the making – knockdown/differentiation) to short-pulse-measured transcription rates at that given point in time. This discrepancy is now discussed in the manuscript as indicated by track-changes. We believe that we have now integrated the observed, albeit moderate, effects on the RARa expression with the hnRNPK/L-ZNF827 model, and that this justifies its inclusion.

3) The biochemical data supporting a complex comprising hnRNP K and L is weak and confusing. hnRNP K can pull down a little bit of L, but hnRNP L pull down does not appear to pull down K (Figure 5E, 6D). This is relevant to the author's main model as the circRNA can efficiently bind L, but hnRNP K is the one that seems to be having the transcriptional control role.

We agree that the results of the co-immunoprecipitations were not entirely clear. Of note, several previous studies have detected a direct interaction between hnRNPK and -L in different cellular settings (e.g. Kim et al., 2000 or Li et al., 2015). Regarding the co-IPs using FLAG-tagged versions shown in Figure 5D, we do observe that the longer isoform of hnRNP L pulls down hnRNP K. Longer exposures of the other mentioned co-IPs (Figure 5E and 6E) indeed show that hnRNP L can co-immunoprecipitate hnRNP K well above IgG control. New and extra panels with longer exposures have been included in Figures 5E and 6E. The reason for less hnRNP K pulldown by hnRNP L, than the other way around, is probably due to the fact that the anti-hnRNP L antibody is less efficient as an IP-grade antibody than the anti-hnRNP K antibody. This will inevitably produce lower signal in the hnRNP K westerns. In addition, both proteins are very abundant with multiple functions and many other interaction partners, which explains that not all of either protein population will interact with the other hnRNP-protein (i.e. stoichiometrically uneven ratio between the proteins depending on the antibody used for immunoprecipitation). In addition, this type of regulation at the transcriptional level, likely does not require many of such hnRNP-circRNA complexes per cell, so the observed “seemingly weak” interactions in these biochemical assays are likely physiologically relevant. Our additional biochemical data using RIP and ChIP also supports this interaction. We believe that the new figures now demonstrate a *bona fide* interaction between hnRNP K and -L, as documented by others before us.

4) Some circRNAs in group 4 (Figure 1E) do not have the same expression pattern upon validation in qPCR (Figure 1G) (see circEZH2, circZFP827). If circZNF827 represses neuronal genes and knocking down circZNF827 upregulates pan neuronal genes (Tubb3, Map2, Nefl and TrkB, Figure 2B), it is counterintuitive to see the circ expression level upregulated during differentiation by RT-qPCR (Figure 1G, the color scale should be replaced to allow proper visualization)?

We agree that the color scaling is poor and this has now been changed. We also agree that the expression patterns are not identical based on RNA-seq and RT-qPCR analyses, respectively (e.g. circEZH2 and circZFP827), which is likely due differences in data normalization. While in the RT-qPCR transformed Ct-values are normalized to GAPDH the heatmap displays RPM-normalized reads (based on CIRI2 algorithm, Materials and methods). These two profiles will become different and as GAPDH also changes slightly upon differentiation (likely slightly changed metabolism in cells starting to exit the cell cycle), resulting in a somewhat skewed picture. Assessing both methods, it is clear that both circZFP827 and circEZH2 become upregulated from mESCs to neuronal progenitor cells (NPCs) and then expression either flattens or goes slightly down at the neuron stage. We do not believe that it is necessarily counterintuitive that an apparent inhibitor of differentiation initially becomes upregulated upon differentiation only to be reduced at the latest stage of this process. *circZNF827* might function to “tap on the brakes” in the beginning of differentiation until certain processes have been correctly initiated and then at a later stage, after commitment to differentiation, become downregulated again. This issue is now included in the discussion. To align the RTqPCR results better with the RNA-seq data, we have chosen another normalization gene, Vinculin, which is more constant. These data are now included and the color scale has been modified for a clearer distinction between circRNAs.

5) The authors showed that there are putative hnRNP-L or K binding sites in circZNF827 (Figure 5—figure supplement 1). Would the deletions or mutations of hnRNP-L or K binding sites in circZNF827 disrupt its interactions with hnRNP-L or K and later the circZNF827-hnRNPK/L-ZNF827 binding to the promoter of NGFR?

Corroborating our models, we have now conducted RNA-immunoprecipitations with hnRNP K/L deletion mutant and found that they no longer efficiently interact with neither hnRNP K nor hnRNP L. Likewise, the rescue experiments described in 1) show that this mutant cannot inhibit the *NGFR* upregulation by knockdown of the endogenous *circZNF827* as expected.